# ARMA Cell: A Modular and Effective Approach for Neural Autoregressive Modeling

**Philipp Schiele**
*Department of Statistics*
*Ludwig-Maximilians-Universität München*

*philipp.schiele@stat.uni-muenchen.de*

**Christoph Berninger**
*Department of Statistics*
*Ludwig-Maximilians-Universität München*

*christoph.berninger@stat.uni-muenchen.de*

**David Rügamer**
*Department of Statistics*
*Ludwig-Maximilians-Universität München*

*david.ruegamer@stat.uni-muenchen.de*

## Abstract

The autoregressive moving average (ARMA) model is a classical, and arguably one of the most studied approaches to model time series data. It has compelling theoretical properties and is widely used among practitioners. More recent deep learning approaches popularize recurrent neural networks (RNNs) and, in particular, long short-term memory (LSTM) cells that have become one of the best performing and most common building blocks in neural time series modeling. While advantageous for time series data or sequences with long-term effects, complex RNN cells are not always a must and can sometimes even be inferior to simpler recurrent approaches. In this work, we introduce the ARMA cell, a simpler, modular, and effective approach for time series modeling in neural networks. This cell can be used in any neural network architecture where recurrent structures are present and naturally handles multivariate time series using vector autoregression. We also introduce the ConvARMA cell as a natural successor for spatially-correlated time series. Our experiments show that the proposed methodology is competitive with popular alternatives in terms of performance while being more robust and compelling due to its simplicity.

## 1 Introduction

Despite the rapidly advancing field of deep learning (DL), linear autoregressive models remain popular for time series analysis among academics and practitioners. Especially in economic forecasting, datasets tend to be small and signal-to-noise ratios low, making it difficult for neural network approaches to effectively learn linear or non-linear patterns. Although research in the past has touched upon autoregressive models embedded in neural networks (e.g., Connor et al., 1991; 1994), existing literature in fields guided by linear autoregressive models such as econometrics mainly focuses on hybrid approaches (see Section 2). These hybrid approaches constitute two-step procedures with suboptimal properties and often cannot even improve over the pure linear model. The DL community took a different route for time-dependent data structures, popularizing recurrent neural networks (RNNs), as well as adaptions to RNNs to overcome difficulties in training and the insufficient memory property (Hochreiter & Schmidhuber, 1997) of simpler RNNs. In particular, methods like the long short-term memory (LSTM) cell are frequently used in practice, whereas older recurrent approaches such as Jordan or Elman networks seem to have lost ground in the time series modeling community (Jordan, 1986; Elman, 1990). This can be attributed to the more stable training and the insensitivity to information lengths in the data of more recent recurrent network approaches such as the LSTM cell.

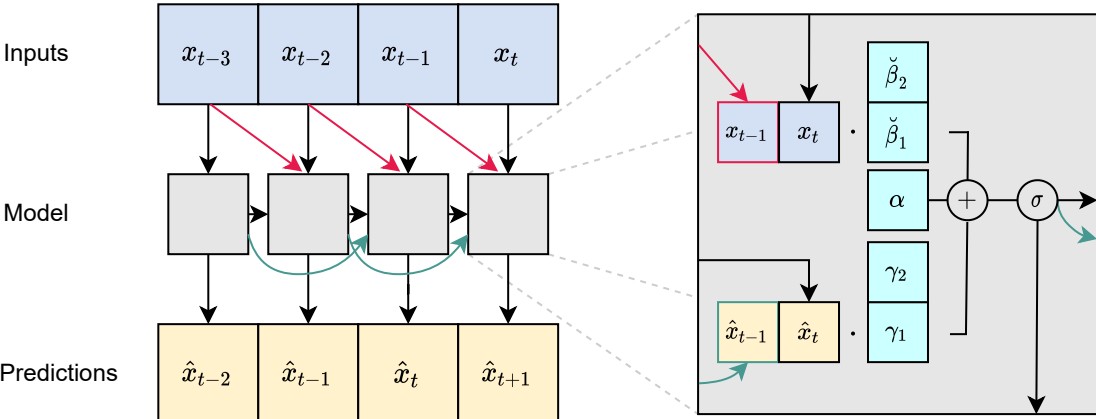

Figure 1: Left: Graphical visualizations of how predictions are computed in a univariate ARMA(2,2) cell using the time series values $x$ from the current and previous time points as well as past model predictions $\hat{x}$. Right: Zooming in on the rightmost model cell from the left picture to show the computations of the ARMA cell with parameters as defined in equation 3.

While often treated as a gold standard, we argue that these more complex RNN cells (such as the LSTM cell) are sometimes used only because of the lack of modular alternatives and that their long-term dependencies or data-driven forget mechanisms might not always be required in some practical applications. For example, in econometrics, including a small number of lagged time series values or lagged error signals in the model is usually sufficient to explain most of the variance of the time series. Similar, sequences of images (i.e., tensor-variate time series) such as video sequences often only require the information of a few previous image frames to infer the pixel values in the next time step(s). In addition, current optimization routines allow practitioners to train classical RNN approaches without any considerable downsides such as vanishing or exploding gradients.

**Our contributions**   In this work, we propose a new type of RNN cell (cf. Figure 1) that can be seen as a natural connection between the classical time series school of thoughts and DL approaches. To analyze how the ARMA modeling philosophy can improve neural network predictions, we

- embed ARMA models in a neural network cell, which has various advantages over classical approaches (see Section 4);

- further exemplify how this proposal can be extended to convolutional approaches to model tensor-variate time series such as image sequences (Section 4.4);

- demonstrate through various numerical experiments that our model is on par with or even outperforms both its classical time series pendant as well as the LSTM cell in various settings; architectures ranging from shallow linear to deep non-linear time series models;

- provide a fully-tested, modular and easy-to-use `TensorFlow` (Abadi et al., 2016) implementation with a high-level syntax almost identical to existing RNN cells to foster its usage and systematic comparisons. It is available at `https://github.com/phschiele/armacell`.

The goal of this paper is further to make practitioners aware of an alternative to commonly used RNN cells, highlight that short-term recurrence can be sufficient in various time series applications, and that a simpler parameterized lag structures can even outperform data-driven forget mechanisms.

We start by discussing related literature in the following section. A short mathematical background is given in Section 3, followed by our proposed modeling approach in Section 4. We investigate practical aspects of our method in Section 5 and summarize all ideas and results in Section 6.

## 2   Related literature

Many advancements in the field of (deep) time series modeling have been made in recent years, in part inspired by autoregressive approaches from statistical modeling (e.g., DeepAR Salinas et al., 2020). As our proposal addresses the connection between classical methods and DL on the level of single building blocks in a network architecture, we focus on literature in classical time series analysis and fundamental RNN modeling approaches in deep learning. More complex architectures and approaches for deep time series modeling can, e.g., be found in Han et al. (2019).

**Traditional autoregressive approaches**   Autoregressive integrated moving average (ARIMA) models (see, e.g., Shumway & Stoffer, 2000) are a general class of linear models for forecasting a time series. They are characterized by a linear function of lags of the dependent variable (response) and lags of the forecasting errors. Boosted by the seminal paper of Box et al. (2015) and further popularized by their simplicity yet practical efficacy, ARIMA models have been extensively studied in statistics, econometrics, and related fields. A special case of ARIMA models are autoregressive moving average (ARMA) models, which are not "integrated", i.e., no differencing steps are required to obtain a stationary mean function. To overcome the linear restrictions of the AR(I)MA model and to account for non-linear patterns observed in real-world problems, several classes of non-linear models have been introduced. Examples include the bilinear model of Granger & Andersen (1978); Rao (1981), the threshold autoregressive model by Tong & Lim (2009), the smooth transition autoregressive model by Chan & Tong (1986), or the Markov switching autoregressive model by Hamilton (2010). Although these approaches have advantages over the linear methods, these models are usually developed to capture very specific non-linear patterns, and hence their generalization is limited. Many common extensions of AR(I)MA models such as seasonal ARIMA (SARIMA) or ARIMA with exogenous variables (ARIMAX) exist (Box et al., 2015). We here only focus on the basic ARMA model, but the proposed approaches in this paper can be easily extended to include the respective peculiarities of real-world time series such as non-stationarity, seasonality, and (non-)linear exogenous variables. Several authors have recognized analogies between ARMA models and RNNs (Connor et al., 1991; 1994; Saxén, 1997), which we will discuss in the following.

**Recurrent neural network approaches**   When modeling sequential data such as time series, RNNs allow previous states to influence the current one, which presents a natural extension of classical perceptron-type connections (Rumelhart et al., 1986). When considering a feedforward neural network with a single hidden layer, such cyclical connections can be established by concatenating the output of the previous time step to the inputs of the next, yielding an Elman network (Elman, 1990). The proposed ARMA cell includes the Elman network as a special case when setting the moving average part of the cell to the value 1. We also demonstrate this in our numerical experiments, showing that both models result in the same parameter estimates. Similarly, a Jordan network is obtained by concatenating the previous hidden layer to the subsequent input (Jordan, 1986). An often-cited shortcoming of these so-called "simple recurrent networks" is their inferiority when learning long-term dependencies, which can be challenging due to vanishing or exploding gradients (e.g., Goodfellow et al., 2016). A variety of methods have since been developed to compensate for this shortcoming, including modeling multiple time scales simultaneously, adding leaky connections or allowing for longer delays within the recurrent connections. Most prominent are gated recurrent units (GRU; Cho et al., 2014) and long short-term memory cells (LSTM; Hochreiter & Schmidhuber, 1997). LSTM cells introduce self-loops that allow the gradient to flow without vanishing even for long durations using an input and a forget gate. GRU cells are similar to LSTM cells but only use a single gating unit to simultaneously control the forgetting and updating mechanism. Both methods have been shown to effectively tackle problems associated with long-term dependencies.

**Combining classical time series approaches with neural networks**   A natural approach to allow for both linear autoregression and flexible non-linearity as provided by RNNs is to combine the two modeling techniques. Various hybrid approaches have been proposed in the past. One of the most common ways to combine the two paradigms is to fit a (seasonal) AR(I)MA model to the time series and subsequently train a neural network to explain the remaining variance in the residuals of the first stage mode (Aslanargun et al., 2007; Fathi, 2019; Tseng et al., 2002; Zhang, 2003). Other approaches specify a (time-delayed) neural network

using the information from a preceding linear model by detrending and deseasonalizing in accordance with the first stage model (Taskaya-Temizel & Ahmad, 2005; Zhang & Qi, 2005).

State space models (SSMs) represent another popular model class in time series analysis (Aoki, 1990). The main advantages of SSMs are their generalized form and possible applications on complex, non-linear time series. A combination of SSMs and RNNs was proposed by Rivals & Personnaz (1996) as well as Suykens et al. (1995) using a network with a state layer between two hidden layers. This yields the so-called state space neural network (Amoura et al., 2011; Zamarreño & Vega, 1998). A more general representation of neural networks as a dynamical system is proposed by Hauser et al. (2019) and a deep combination of SSMs and RNNs was further suggested by Rangapuram et al. (2018). While it is possible to represent ARMA models using SSMs, existing neural network SSM approaches in the literature do not aim for a general and modular building block, but propose specific networks using a fixed architecture.used for the mapping from features to SSM parameters.

Error Correction Models (ECMs) are commonly used to forecast cointegrated time series, i.e., time series that have a common long-term stochastic trend. For such time series, the information contained in the levels of the data is lost during a differencing step, making ECMs more suitable compared to ARIMA models. Mvubu et al. (2020) introduced a neural variant of ECMs, the Error Correction Neural Networks.

While some hybrid methods such as the combination of graph neural networks with an LSTM network (Smyl, 2020) have been shown to excel in time series forecasts, the efficacy of hybrid approaches combining AR(I)MA models and RNNs is often not clear (Taskaya-Temizel & Ahmad, 2005; Terui & Van Dijk, 2002). Khashei & Bijari (2011) propose a two-stage procedure that is guaranteed to improve the performance of the two single models, but as for all other existing hybrid approaches their technique requires fitting two separate models and hence cannot be combined or extended in a straightforward manner.

**Recurrent convolutional approaches**  For spatio-temporal sequence forecasting problems, an extension to fully-connected RNN cells are RNN cells that apply convolutional operations to the spatially distributed information of multiple time series. As for time series applications, popular approaches use long-memory mechanisms, e.g., convolutional GRU adaptations (Tian et al., 2019) or the ConvLSTM (Shi et al., 2015) as an extension of the LSTM cell for spatio-temporal data.

## 3  Background and notation

In the following, we introduce our notation and the general setup for modeling time series. We will address univariate time series $x_t \in \mathbb{R}$ for time points $t \in \mathbb{Z}$ as well as multi- and tensor-variate time series, which we denote as $\boldsymbol{x}_t$ and $\boldsymbol{X}_t$, respectively.

**ARMA model**  The ARMA$(p, q)$ model (Box et al., 2015) for $p, q \in \mathbb{N}_0$ is defined as

$$x_t = \alpha + \sum_{i=1}^{p} \beta_i x_{t-i} + \sum_{j=1}^{q} \gamma_j \varepsilon_{t-j} + \varepsilon_t, \tag{1}$$

where $x_t$ represents the variable of interest defined for $t \in \mathbb{Z}$ and is observed at time points $t = 1, \ldots, T$, $T \in \mathbb{N}^1$. $\alpha, \beta_1, \ldots, \beta_p, \gamma_1, \ldots, \gamma_q$ are real valued parameters and $\varepsilon_t \overset{iid}{\sim} \mathcal{F}(\sigma^2)$ is an independent and identically distributed (iid) stochastic process with pre-specified distribution $\mathcal{F}$ and variance parameter $\sigma^2 > 0$. By setting $q = 0$ or $p = 0$, the ARMA model comprises the special cases of a pure autoregressive (AR) and a pure moving average (MA) model, respectively. The class of ARMA models is, in turn, a special case of ARIMA models, where differencing steps are applied to obtain a stationary mean function before fitting the ARMA model. As stationarity is also a fundamental assumption for RNNs to justify parameter sharing (Goodfellow et al., 2016), we focus on the class of ARMA models in this work, i.e., assume that differencing has already been applied to the data. series is characterized by a constant mean and variance and a time invariant

---

[1]It is common to define a time series for time points $t = 1, \ldots, T$ to describe its current value and recent history, while time series dynamics are assumed to originate from time points prior to $t = 1$, hence $t \in \mathbb{Z}$

autocorrelation structure. $\alpha_0, \dots, \alpha_p$ and $\beta_1, \dots, \beta_q$ are model parameters and $p$ and $q$ characterize the number of lags of the dependent variable and the forecasting errors included in the model, respectively.

**VARMA model** The univariate ARMA model can be generalized to a multivariate version – the vector autoregressive moving average (VARMA) model – by adapting the principles of the ARMA model for multivariate time series. The VARMA$(p, q)$ model (Tiao & Box, 1981) for $p, q \in \mathbb{N}_0$ is defined as

$$\boldsymbol{x}_t = \boldsymbol{\alpha} + \sum_{i=1}^{p} \boldsymbol{B}_i \boldsymbol{x}_{t-i} + \sum_{j=1}^{q} \boldsymbol{\Gamma}_j \boldsymbol{\varepsilon}_{t-j} + \boldsymbol{\varepsilon}_t \tag{2}$$

where $\boldsymbol{x}_t, t \in \mathbb{Z}$ represents a vector of time series observed at time points $t = 1, \dots, T$. $\boldsymbol{B}_i$ and $\boldsymbol{\Gamma}_j$ are time-invariant $(k \times k)$-matrices, where $k \in \mathbb{N}$ represents the number of individual time series. $\boldsymbol{\varepsilon}_t$ is a $k$-dimensional iid stochastic process with pre-specified $k$-dimensional distribution $\mathcal{F}(\boldsymbol{\Omega})$ and covariance matrix $\boldsymbol{\Omega}$. By setting $q = 0$, the VARMA model comprises the special cases of a pure autoregressive (VAR) model, which is the most common VARMA model used in applications. Similar to the ARMA model being a special case of the ARIMA model class, the VARMA model is a special case of the VARIMA model class, representing only stationary time series.

## 4 ARMA-based neural network layers

ARMA models have been successfully used in many different fields and are a reasonable modeling choice for time series in many areas. This section introduces a neural network cell version of the ARMA mechanism. While very similar to Elman or Jordan networks, the proposed cell exactly resembles the ARMA computations and can be used in a modular fashion in any neural network architecture where recurrent structures are present. Emulating the ARMA logic in a recurrent network cell has various advantages. It allows to 1) recover estimated coefficients of classical ARMA software (see Supplementary Material B.1 for an empirical investigation of the convergence), but can also be used to fit ARMA models for large-scale or tensor-variate data (which is otherwise computationally infeasible), 2) modularly use the ARMA cell in place for any other RNN cell, 3) combine ARMA approaches with other features from neural networks such as regularization and thereby seamlessly extend existing time series models, and 4) model hybrid linear and deep network models that were previously only possible through multi-step procedures. As shown in our numerical experiments section, an ARMA cell can further lead to comparable or even better prediction performance compared to modern RNN cells.

### 4.1 ARMA cell

An alternative formulation of the ARMA model can be derived by incorporating the observed (or estimated) residual term $\hat{\varepsilon}$ through the predictions $\hat{x}_t \coloneqq x_t - \hat{\varepsilon}_t, t \in \mathbb{Z}$. Thus, equation 1 can be defined in terms of its intercept, the model predictions $\hat{x}_t$ and the actual time series values $x_t$. It follows:

$$\hat{x}_t = \alpha + \sum_{i=1}^{\max(p,q)} \breve{\beta}_i x_{t-i} - \sum_{j=1}^{q} \gamma_j \hat{x}_{t-j} \quad \text{with } \breve{\beta}_i = \begin{cases} \beta_i + \gamma_i & \text{for } i \leq \min(p, q), \\ \beta_i & \text{for } i > q \text{ and } p > q, \\ \gamma_i & \text{for } i > p \text{ and } p < q. \end{cases} \tag{3}$$

Using equation 3, we can implement the ARMA functionality as an RNN cell. More specifically, the recurrent cell processes the $p$ lagged time series values as well as the $q$ predicted outputs of the previous time steps and computes a linear combination with parameters $\breve{\beta}_i$ and $\gamma_j$. After adding a bias term, the final output $\hat{x}_t$ is given by a (non-linear) activation function $\sigma$ of the sum of all terms. Figure 1 gives both a higher-level view of how predictions are computed in the ARMA cell as well as a description of how the cell is defined in detail. In addition to the classical ARMA computations in the cell, the activation function $\sigma$ allows to straightforwardly switch between a linear ARMA model and a non-linear version.

The above-mentioned ARMA cell has the same hypothesis space as the classical ARMA model when using a single-unit ARMA cell with a linear activation function. While using a non-linear activation for the outputs,

in this case, is equivalent to using a link function (as done in generalized linear models) for the classical ARMA model, extensions using multiple units or stacking ARMA cells (see below) increase the model's expressiveness. As for regular multi-layer perceptrons, where each node is a simple regression model with an activation function and the combination of multiple units makes the models more expressive, these extensions combine simpler ARMA models and therefore allow modeling more complex relationships.

**Advantages and comparison to other cells**  Modeling a classical ARMA model in a neural network can be more stable in the estimation of coefficients due to the use of a stochastic first-order method (less vulnerable to ill-conditioning and numerical instabilities), which is also confirmed in our numerical experiments in numerous settings. Training the ARMA model using mini-batch optimization, further allows scaling to large data sets, which is especially beneficial when modeling multivariate time series where the complexity of classical models increases substantially with the number of parameters and the multivariate time series dimension.

In contrast to the standard RNN cell, the ARMA cell internally can access multiple previous states and lagged features, making it potentially easier to learn time dependencies and recurrences. The standard RNN cell, in contrast, only relies on the current input and the previous cell state. In other words, the ARMA cell allows for a more complex autoregressive structure and, in contrast to the simple RNN, provides a way to model moving averages. This can also be explained using Figure 1, where the standard RNN cell can represent the black arrows, but not the red and green connections.

Last but not least, the ARMA cell can be used to seamlessly model hybrid models end-to-end in one holistic network, which historically has always been implemented using two-step approaches (Zhang, 2003), yielding potentially inferior performance as the models in both steps are not jointly optimized. This is also confirmed by our numerical results in Section 5.3.

## 4.2   Training procedure

The ARMA cell is trained as follows. For a given sequence, it creates predictions by recursively applying equation 3. This is done in one forward pass. To also allow predictions for the first $q$ time points in each sequence, we need to pad the sequence of previous predictions with 0-values. Further details on the input sizes of the ARMA cell can be found in Supplementary Material D.3. We then differentiate the loss of these outputs given the current weights back through the whole sequence, i.e., the network is trained exactly as done for the LSTM, GRU, and simple RNN cell via backpropagation through time. Note that our ARMA cell also supports returning sequences, which we can use to stack cells or for training a model on multiple steps simultaneously.

## 4.3   Extensions

The ARMA cell in Figure 1 can be used in a modular fashion similar to an LSTM or GRU cell. In the following, we will thus present how this idea can be used to generate more complex architectures using multiple units or by stacking cells. Both options also allow bypassing the linearity assumptions of ARMA models.

**Multi-unit ARMA cell**  Similar to feedforward neural networks, an RNN layer can also contain multiple units. Each unit receives the same input but can capture different effects due to the random initialization of weights. The outputs of each unit are then concatenated. In the left panel of Figure 2, a multi-unit architecture allows combining different activation functions to simultaneously capture linear and non-linear effects. Using a multi-unit ARMA cell thereby seamlessly provides the possibility to combine a linear with a non-linear ARMA model. We refer to models having a single hidden ARMA layer with one or more units as *ShallowARMA* models.

**Stacked ARMA**  To allow for higher levels of abstraction and increased model complexity, the ARMA modeling strategy does not only allow for multiple units in a single layer, but users can also stack multiple layers in series. This is achieved by returning a sequence of lagged outputs from the previous layer, as

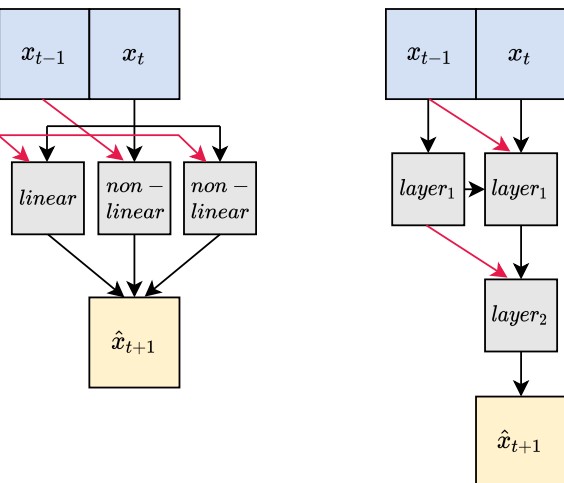

Figure 2: Visualization of an ARMA cell with multiple units representing a mixture of linear and non-linear ARMA models by using different activation functions (left) and a network with stacked ARMA cells creating a more complex model class by transforming inputs by subsequent ARMA cells (right).

depicted on the right of Figure 2. Models with more than one hidden ARMA layer are referred to as *DeepARMA* models in the following.

### 4.4 ConvARMA

Similar to the ConvLSTM network (Shi et al., 2015), it is possible to model spatial dependencies and process tensor-variate time series $\boldsymbol{X}_t \in \mathbb{R}^{n_1 \times \ldots \times n_d}, n_1, \ldots, n_d \in \mathbb{N}, d \in \mathbb{N}$ by using convolution operations within an ARMA cell. The resulting ConvARMA$(p, q)$ cell for $p, q \in \mathbb{N}_0$ and $t \in \mathbb{Z}$ is defined as

$$\boldsymbol{I}_t = \sum_{i=1}^{p} \boldsymbol{W}_i * \boldsymbol{X}_{t-i}, \quad \boldsymbol{C}_t = \sum_{j=1}^{q} \boldsymbol{U}_j * \hat{\boldsymbol{X}}_{t-j}, \quad \hat{\boldsymbol{X}}_t = \sigma \left( \boldsymbol{I}_t + \boldsymbol{C}_t + \boldsymbol{b} \right), \tag{4}$$

where $*$ represents the convolution operator, $\boldsymbol{W}_i \in \mathbb{R}^{k_1 \times \ldots \times k_{d-1} \times n_d \times c}, i = 1, \ldots, p$ and $\boldsymbol{U}_j \in \mathbb{R}^{k_1 \times \ldots \times k_{d-1} \times c \times c}, j = 1, \ldots, q$ are the model's kernels of size $k_1 \times \ldots \times k_{d-1}$, $\boldsymbol{b} \in \mathbb{R}^c$ is a bias term broadcasted to dimension $n_1 \times \ldots \times n_{d-1} \times c$ and $\sigma$ an activation function. By convention, the last dimension of the input represents the channels, and $c$ denotes the number of filters of the convolution. The inputs of the convolution are padded to ensure that the spatial dimensions of the prediction $\hat{\boldsymbol{X}}_t$ and the state remain unchanged. In other words, the ConvARMA cell resembles the computations of an ARMA model, but instead of simple multiplication of the time series values with scalar-valued parameters, a convolution operation is applied. Figure 3 shows an abstract visualization of the computations in a ConvARMA cell. To follow the AR(I)MA modeling logic in the spatial dimensions, a ConvARMA cell can further incorporate spatial differences in all directions. A possible extension of the cell proposed in equation 4 could further be to allow for non-linear recurrent activations as done for, e.g., the ConvLSTM cell.

As for the ConvLSTM or ConvGRU cell, the ConvARMA cell can be included in an autoencoder architecture for sequence-to-sequence modeling or extended to e.g., allow for warping, rotation, and scaling (Shi et al., 2017).

### 4.5 Limitations

As for other autoregressive approaches, our approach is limited in its application if the time series are very short or if a large number of lags $p$ is required to approximate the underlying data generating process well. We note, however, that due to the model's recurrent definition, past time points $t - i$ for $i > p$ also influence the model's predictions. It is therefore often not necessary to define a large lag value $p$, even if autocorrelation is

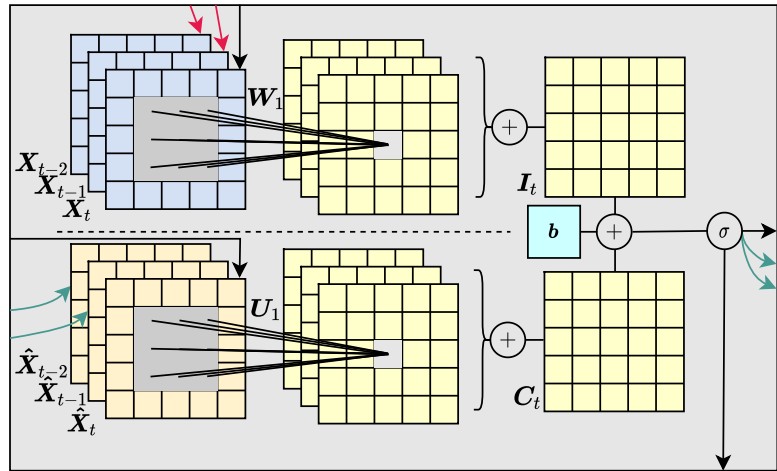

Figure 3: Exemplary visualization of a single-filter ConvARMA cell processing matrix-variate time series (with a single channel) with three lags (upper left) and matrix-variate predictions with three lags (bottom left) using convolutions and combining the results into a single matrix prediction (bottom/top right) with additional bias term $b$ and activation function $\sigma$ (center right).

high. Despite the ARMA cell's simplicity, this also shows that its predictions are not always straightforward to interpret.

## 5   Numerical experiments

In this section, we examine the performance of our ARMA cell in a variety of synthetic and benchmark experiments. We examine how it compares to classical time series approaches as well as to similar complex neural architectures. Note that our experiments are not designed to be a benchmark comparison with current state-of-the-art time series forecasting frameworks. These rather complex architectures include many different components, such as automated pre-processing and feature generation, and thus do not necessarily allow making a statement about the performance of a single recurrent cell therein. Instead, we aim for a comparison with other fundamental modeling building blocks [2]. Yet, in order to emphasize our cell's modularity and demonstrate its efficacy when used as part of a larger state-of-the-art network, we also present results on real-world benchmark data sets when replacing RNN cells within a DeepAR model (Salinas et al., 2020) with the ARMA cell.

**Methods**   For (multivariate) time series, we compare a shallow and a deep variant of the ARMA cell against the respective (V)ARMA model and neural models. For the latter, we consider LSTM, GRU, and Simple RNN cells, again each in a their shallow and deep variants. Hyperparameter optimization is done using a grid search with predefined parameter spaces for the number of units for all network layers and lags for ARMA-type models. All other hyperparameters of network layers are kept fixed with defaults that do not favor one or the other method. For moving images, we compare ConvARMA against a naïve approach of repeating the last image and a ConvLSTM network with parameter specifications that are defined as similar as possible to the one of ConvARMA. Further details on the specification of the architectures can be found in the Supplementary Material B.6.

**Performance measures**   We compare time series predictions using the root mean squared error (RMSE) for uni- and multivariate time series forecasts, and the cross-entropy loss for next frame video predictions. We provide further performance measures for our comparisons in the Supplementary Material E.

---

[2]We provide code to reproduce all experiments at `https://github.com/phschiele/armacell_paper`

Table 1: Comparisons of different methods (rows) and different data generating processes (columns) for univariate time series using the average RMSE ± the standard deviation of 30 independent runs. The best performing method is highlighted in bold, the second-best in italics.

| MODEL | ARMA | TAR | SGN | NAR | HETEROSKEDASTIC |
|---|---|---|---|---|---|
| ARMA | 2.04±0.35 | 2.92±3.20 | 2.36±2.67 | 2.67±4.92 | 1.19±0.15 |
| SHALLOWARMA | **1.96±0.09** | **1.09±0.12** | 1.10±0.07 | *1.02±0.05* | **1.11±0.06** |
| DEEPARMA | 1.97±0.10 | *1.24±0.47* | **1.05±0.06** | 1.02±0.04 | *1.11±0.06* |
| LSTM | 1.98±0.10 | 1.38±0.42 | 1.19±0.16 | 1.02±0.04 | 1.15±0.09 |
| DEEPLSTM | 2.02±0.10 | 1.46±0.55 | 1.17±0.13 | 1.02±0.04 | 1.16±0.08 |
| GRU | *1.96±0.10* | 1.28±0.31 | *1.09±0.08* | **1.02±0.04** | 1.13±0.07 |
| DEEPGRU | 1.99±0.09 | 1.24±0.36 | 1.09±0.12 | 1.02±0.04 | 1.12±0.06 |
| SIMPLE | 1.99±0.09 | 1.29±0.31 | 1.14±0.09 | 1.04±0.04 | 1.13±0.08 |
| DEEPSIMPLE | 2.01±0.11 | 1.47±0.57 | 1.15±0.10 | 1.03±0.04 | 1.16±0.10 |

**Comparison between classical and first-order optimization**   In the case where the data generating process is in fact a (V)ARMA process, we expect the classical (V)ARMA model and the ARMA cell to perform similarly, but note that the optimization using stochastic gradient descent can sometimes yield better estimations of this process and hence outperform these classical models despite having the exact same hypothesis space.

## 5.1   Simulation study

We start with a variety of synthetic data examples using time series models defined in Lee et al. (1993). Simulations include linear and non-linear, as well as uni- and multivariate time series. All time series are of length 1000 and split into 70% train and 30% test data. The data generating processes follow Lee et al. (1993) and include an ARMA process (ARMA), a threshold autoregressive model (TAR), an autoregressive time series which is transformed using the sign operation (SGN), a non-linear autoregressive series (NAR), a heteroscedastic MA process (Heteroscedastic), a vector ARMA (VARMA), a non-linear multivariate time series with quadratic lag structure (SQ), and an exponential multivariate autoregressive time series (EXP). The exact specification of the data generating processes can be found in the Supplementary Material B.3.

**Results**   The results in Table 1 suggest that the ShallowARMA approach emulating an ARMA model in a neural network works well for all linear- and non-linear datasets. In terms of robustness, the lower RMSE and high standard deviation of the ARMA model on the ARMA process shows that fitting an ARMA model in a neural network with stochastic gradient descent can, in fact, be more robust than the standard software (Hyndman & Khandakar, 2008; Seabold & Perktold, 2010). While the classical ARMA did match the performance of its neural counterpart it some cases, the average RMSE is worse, as it did not converge in all runs, even for the linear time series. The performance of the DeepARMA approach is slightly worse compared to the ShallowARMA in most cases. The performance of LSTM, GRU, and the Simple RNN are all similar, with all methods matching the ARMA cells in some cases, and falling slightly behind in others. As expected, the classical ARMA approach does not work well for non-linear data generating processes (TAR, SGN, NAR) and yields unstable predictions underpinned by the large standard deviations in RMSE values.

For multivariate time series results of the simulation are summarized in Table 2. The results again show that the ShallowARMA model matches the performance of the classical VARMA model for a dataset that is also based on a VARMA process. For other types of data generation, the ShallowARMA model and DeepARMA model work similarly well. Both outperform the other neural cells, which in turn yield better results than the VARMA baseline.

In summary, findings suggest that ARMA cells work well for simpler linear and non-linear data generating processes while being much more stable than a classical ARMA approach. In Supplementary Material B.1, we further study the empirical convergence of a single unit single hidden layer ARMA cell, which is mathe-

Table 2: Comparisons of different methods (rows) and different data generating processes (columns) for multivariate time series using the average RMSE ± the standard deviation of 30 independent runs. The best performing method is highlighted in bold, the second-best in italics.

|  | VARMA | EXP | SQ |
|---|---|---|---|
| VARMA | **1.00±0.03** | 3.35±0.69 | 1.90±0.14 |
| SHALLOWARMA | *1.00±0.03* | *3.14±0.74* | **1.75±0.12** |
| DEEPARMA | 1.01±0.03 | **3.10±0.75** | *1.76±0.11* |
| LSTM | 1.01±0.04 | 3.26±0.73 | 1.83±0.16 |
| DEEPLSTM | 1.03±0.04 | 3.35±0.68 | 1.86±0.14 |
| GRU | 1.02±0.04 | 3.19±0.75 | 1.80±0.13 |
| DEEPGRU | 1.01±0.04 | 3.22±0.80 | 1.82±0.18 |
| SIMPLE | 1.02±0.03 | 3.29±0.70 | 1.80±0.12 |
| DEEPSIMPLE | 1.02±0.03 | 3.30±0.84 | 1.83±0.13 |

matically equivalent to an ARMA model for given values of $p$ and $q$, and present another comparison showing the equivalence of the ARMA cell and an Elman network in Supplementary Material B.2.

## 5.2  Ablation studies

To explore the validity of our presented results, we perform a series of ablation studies. Two important influence factors on the performance of time series models are the length of the time series and the forecasting horizon, which we subsequently assess in the controlled setting of our simulation study.

### 5.2.1  Time series length

In order to investigate the influence of the time series length on the performance reported in previous simulations, we vary $T \in \{200, 1000, 10000\}$ and re-run the experiments reported in Table 1 and 2. The results are given in Tables 7 and 8 in Supplementary Material B.4. In summary, the rank of the different methods is similar to the aforementioned results, and ShallowARMA yields the best results in most settings. There is, however, a clear trend in that the performance differences between the different cell types become irrelevant for an increase in $T$. For example, for the multivariate time series study setting SQ, the ShallowARMA yields a notably better MSE for $T = 200$ compared to the DeepSimple cell ($1.97 \pm 0.41$ vs. $3.00 \pm 3.95$), the performances are almost identical for $T = 10,000$ observations ($1.78 \pm 0.05$ vs. $1.80 \pm 0.07$).

### 5.2.2  Forecasting horizon

Similar to the previous ablation study, we reran the experiments but now alter the forecasting horizon by comparing a one-step, 10-step, and 20-step forecast for $T = 1000$. The results can be found in Table 9 and 10 in Supplementary Material B.4. In the univariate case for forecasting horizons greater one, the different ARMA variations do not outperform other approaches anymore and DeepLSTM, GRU, or DeepGRU yield the best results in many cases. The performance values, however, are in most cases within one standard deviation of those by the Shallow- or DeepARMA approach. For the multivariate case, the classical VARMA model provides the best forecast for all multi-step ahead forecast scenarios, closely followed by the Shallow- and DeepARMA models.

## 5.3  Comparison to hybrid models

We now investigate the differences between a standard hybrid approach following Zhang (2003) and an end-to-end approach using the ARMA cell. The hybrid model first trains a classical model, in this case, an ARMA(2,2) model. Then, an LSTM model is fit on the residual. The final prediction is obtained as the sum of the ARMA and LSTM predictions. We also implement an end-to-end version of this model using the ARMA cell, which we refer to as End2End. Here, we train a linear ARMA cell, also specified with $p = 2$ and $q = 2$, and sum its output with the output from an LSTM cell. Both model approaches have the

|          | ARMA               | TAR                | SGN                | NAR                | Heteroskedastic    |
| -------- | ------------------ | ------------------ | ------------------ | ------------------ | ------------------ |
| End2End  | $2.021 \pm 0.105$  | $\mathbf{1.134 \pm 0.127}$ | $\mathbf{1.128 \pm 0.055}$ | $\mathbf{1.002 \pm 0.044}$ | $\mathbf{1.142 \pm 0.062}$ |
| Hybrid   | $\mathbf{2.014 \pm 0.112}$ | $1.264 \pm 0.811$  | $1.138 \pm 0.051$  | $3.009 \pm 8.869$  | $1.147 \pm 0.062$  |

Table 3: Comparisons of the End2End and Hybrid approach on different data generating processes (columns) for univariate time series using the average RMSE $\pm$ the standard deviation of 30 independent runs. The best-performing method is highlighted in bold.

same hypothesis space, however, training a single model simplifies the training process and optimized the parameters jointly. We run both approaches on the simulated univariate time series, as shown in Table 3. We see that the End2End model performs similarly to the hybrid model in most cases. However, for the NAR dataset, we find that the two-step hybrid approach does not converge in all cases, leading to a substantially worse average RMSE and a high corresponding standard deviation, indicating that this approach is less robust compared to the End2End model.

## 5.4 Benchmarks

In order to investigate the performance of our approach for real-world time series with a potentially more complex generating process, we compare the previously defined models on various time series benchmark datasets.

### 5.4.1 Univariate and multivariate time series

We use the m4 (Makridakis et al., 2018), traffic (Yu et al., 2016), electricity (Yu et al., 2016) and exchange (Lai et al., 2018) dataset, all openly accessible and commonly used in time series forecast benchmarks. Further background on every dataset and details on pre-processing can be found in the Supplementary Material B.5. As all datasets come with multiple time series, we use these datasets both for testing the performance on univariate and multivariate time series. For univariate time series, this is done by training a local model for every dimension and averaging the results over the different multivariate dimensions. The multivariate comparison is based on the predictions of a single global model.

**Univariate time series**   Results of univariate benchmarks are summarized in Table 4. The comparisons suggest that the two ARMA cells and the other neural cells perform equally well on the Exchange dataset, but the ARMA cells outperform on the other datasets. The classical ARMA model is competitive for the m4 dataset, but again worse than its neural pendant on Traffic, Electricity, and Exchange.

**Multivariate time series**   For the multivariate time series benchmarks, we observe that model performance is in general worse than when performing hyperparameter optimization and model training for each time series individually, as done for the univariate time series benchmark. Finding architectures better suited to the individual time series seems to outweigh the additional information from observing the comovement of multiple time series simultaneously. In the comparison of different forecasting approaches for multivariate dimensions, the performance of the ARMA cells is either notably better than the other neural cells but on par with the classical ARMA model (Traffic), better than the ARMA model but on par with the other neural cells (Exchange), or outperforms all other approaches (m4). Only for the Electricity dataset, the ARMA cells yield a slightly worse MSE compared to the DeepSimple cell.

### 5.4.2 Integration with state-of-the-art forecasting frameworks

To demonstrate the modularity aspect of the ARMA cell, we use it to replace the LSTM cell in a DeepAR model (Salinas et al., 2020). We then train a larger global model on the previously studied benchmark data sets for multivariate time series (as this is the application area where DeepAR model excels in performance) and examine to what extent the change in RNN cell influences the results. The experimental details can be found in Supplementary Material D.4.3. Results (Table 5) indicate that it is possible to successfully replace

Table 4: Comparison of different univariate and multivariate forecasting approaches (rows) for different datasets (columns) based on the average RMSE ± the standard deviation of 10 independent runs. The best performing method is highlighted in bold, the second-best in italics.

| | | M4 | TRAFFIC | ELECTRICITY | EXCHANGE |
|---|---|---|---|---|---|
| UNIV. | ARMA | 1.58±0.00 | 0.98±0.00 | 1.19±0.00 | 1.18±0.00 |
| | SHALLOWARMA | **1.57±0.01** | 0.97±0.00 | 1.14±0.01 | *1.03±0.00* |
| | DEEPARMA | *1.57±0.01* | **0.94±0.01** | **1.10±0.02** | 1.03±0.00 |
| | LSTM | 1.71±0.14 | *0.96±0.01* | 1.15±0.07 | 1.03±0.01 |
| | DEEPLSTM | 1.96±0.38 | 0.97±0.02 | 1.12±0.05 | 1.04±0.01 |
| | GRU | 1.61±0.03 | 0.97±0.02 | *1.11±0.02* | 1.04±0.01 |
| | DEEPGRU | 1.61±0.02 | 0.97±0.02 | 1.11±0.03 | 1.04±0.00 |
| | SIMPLE | 1.72±0.15 | 1.00±0.01 | 1.12±0.01 | 1.04±0.00 |
| | DEEPSIMPLE | 1.76±0.17 | 1.00±0.02 | 1.11±0.02 | 1.04±0.01 |
| MULTIV. | ARMA | 1.72±0.00 | *1.06±0.00* | 1.46±0.00 | 1.33±0.00 |
| | SHALLOWARMA | *1.68±0.01* | **1.06±0.00** | 1.37±0.03 | *1.10±0.00* |
| | DEEPARMA | **1.67±0.01** | 1.08±0.01 | 1.32±0.03 | 1.10±0.00 |
| | LSTM | 1.92±0.15 | 1.15±0.01 | 2.07±1.12 | **1.10±0.01** |
| | DEEPLSTM | 2.11±0.25 | 1.15±0.02 | 1.26±0.05 | 1.18±0.25 |
| | GRU | 1.91±0.25 | 1.15±0.00 | 1.25±0.04 | 1.10±0.00 |
| | DEEPGRU | 1.88±0.12 | 1.15±0.01 | *1.23±0.02* | 1.10±0.00 |
| | SIMPLE | 1.89±0.06 | 1.16±0.00 | 1.25±0.03 | 1.10±0.00 |
| | DEEPSIMPLE | 1.90±0.08 | 1.16±0.01 | **1.22±0.01** | 1.10±0.00 |

Table 5: Comparison of different DeepAR-based models (rows) for different datasets (columns) based on the average negative log-likelihood ± its standard deviation of 30 independent runs. The best performing method is highlighted in bold.

| | M4 | TRAFFIC | ELECTRICITY | EXCHANGE |
|---|---|---|---|---|
| DEEPAR ARMA SINGLE | 2.444 ± 0.137 | **1.323 ± 0.033** | **5.236 ± 0.398** | 1.552 ± 0.025 |
| DEEPAR LSTM SINGLE | **2.306 ± 0.056** | 1.362 ± 0.054 | 10.236 ± 9.242 | **1.510 ± 0.034** |
| DEEPAR ARMA STACKED | 2.513 ± 0.157 | **1.332 ± 0.056** | **5.639 ± 0.461** | 1.551 ± 0.026 |
| DEEPAR LSTM STACKED | **2.266 ± 0.049** | 1.381 ± 0.053 | 9.607 ± 5.630 | **1.494 ± 0.033** |

the LSTM with an ARMA cell in the DeepAR model and to receive a similar performance. We further observe that the LSTM-based DeepAR does not always converge for the electricity dataset, indicating that the training of the ARMA cell is more robust.

### 5.4.3 Tensor-variate time series

Finally, we compare the ARMA and LSTM approach on tensor-variate time series such as image or sensor-grid sequences. These benchmarks are performed with different layers and filter sizes to investigate the difference in performance for different RNN cell complexity. As a baseline model, we report the performance of simply predicting the image of the previous frame.

**Datasets** To compare the models' performance, we use five different datasets. A common dataset for next video frame prediction is the Moving MNIST dataset (MovMNIST; Srivastava et al., 2015) which contains video sequences of two digits moving randomly inside a frame. Similarly, the Noisy and Shifted squares datasets (Noisy, Shifted) used to investigate the properties temporal convolutions (Chollet et al., 2015) consist of smaller and bigger squares moving through a pre-defined window at different speeds. In addition, we analyze two spatio-temporal datasets, the Taxi NYC and Bike NYC datasets (NYTaxi, NYBike; as, e.g., used in Lin et al., 2020). These consist of hourly taxi and bike movements in New York City quantified as the number of inflows and outflows of each sector in a grid view of the city. A more detailed description of all datasets can be found in Supplementary Material B.5.

Table 6: Comparison of different forecasting approaches (rows; with numbers corresponding to the quadratic filter sizes of each layer) for different datasets (columns) based on the average cross-entropy (standard deviation in brackets) over 10 different initializations. The best-performing method is highlighted in bold.

|  | MovMNIST | Noisy | Shifted | NYTaxi | NYBike |
|---|---|---|---|---|---|
| ConvARMA 5-3-1 | **0.063±0.001** | **0.094±0.002** | **0.082±0.002** | **0.281±0.001** | **0.285±0.001** |
| ConvLSTM 5-3-1 | 0.076±0.038 | 0.116±0.002 | 0.109±0.004 | 0.285±0.001 | 0.290±0.002 |
| ConvARMA 3-1 | **0.072±0.003** | **0.149±0.057** | **0.151±0.065** | **0.288±0.000** | **0.292±0.000** |
| ConvLSTM 3-1 | 0.093±0.002 | 0.161±0.002 | 0.154±0.002 | 0.289±0.000 | 0.295±0.000 |
| ConvARMA 3 | **0.075±0.000** | **0.120±0.002** | **0.112±0.001** | 0.289±0.000 | **0.296±0.000** |
| ConvLSTM 3 | 0.103±0.018 | 0.167±0.002 | 0.159±0.001 | **0.289±0.000** | 0.296±0.001 |
| Baseline | 0.509 | 1.041 | 1.135 | 0.375 | 0.391 |

**Results**  Table 6 summarizes the comparisons of tensor-variate forecasts. Similar to the univariate and multivariate time series applications, the tensor-variate version of the ARMA cell outperforms its LSTM pendant in all configurations on Moving MNIST as well as the noisy and shifted datasets. For the Taxi and Bike data, both methods perform on par while being notably better than the baseline.

## 6   Conclusion and Outlook

We provided a modular and flexible neural network cell to model time series in a simply parameterized fashion and as an alternative to commonly used RNN cells such as the LSTM cell. We further extended this approach to vector autoregression and autoregressive models for tensor-variate applications. Our numerical experiments show that the ARMA cell 1) performs well on univariate, multivariate, and tensor-variate time series; 2) matches or even outperforms the LSTM, GRU, and a Simple RNN cell in linear and non-linear settings, and; 3) shows more robust convergence for classical ARMA formulations compared to a standalone implementation.

**Outlook**  As noted by an anonymous reviewer, an interesting future research direction is to make use of the theoretical results for ARMA models known from classical statistical literature and transfer these to the application of ARMA as a cell with multiple units or in its stacked variant. A directly available result, e.g., would be last-layer uncertainty quantification (see, e.g., Immer et al., 2021) in a stacked RNN model where the last cell is an ARMA cell with one unit. Although this neglects the variance in previous layers, it allows a first assessment of the RNN's uncertainty. Further, when merging multiple linearly activated ARMA cells, the combination is an ensemble of ARMA models, for which some form of uncertainty quantification method could be derived.

## Acknowledgements

This work has been funded by the German Federal Ministry of Education and Research and the Bavarian State Ministry for Science and the Arts. The authors of this work take full responsibility for its content.

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

## A   Derivation ARMA formula

This shows how to rewrite the ARMA model. We start with

$$x_t = \alpha + \sum_{i=1}^{p} \beta_i x_{t-i} + \sum_{j=1}^{q} \gamma_j \varepsilon_{t-j} + \varepsilon_t$$

and use the definition of $\hat{x}_t \coloneqq x_t - \varepsilon_t$ to get

$$\hat{x}_t = \alpha + \sum_{i=1}^{p} \beta_i x_{t-i} + \sum_{j=1}^{q} \gamma_j \varepsilon_{t-j}.$$

We now replace each $\varepsilon_{t-j}$ with $x_{t-j} - \hat{x}_{t-j}$

$$\hat{x}_t = \alpha + \sum_{i=1}^{p} \beta_i x_{t-i} + \sum_{j=1}^{q} \gamma_j (x_{t-j} - \hat{x}_{t-j}) = \alpha + \sum_{i=1}^{p} \beta_i x_{t-i} + \sum_{i=1}^{q} \gamma_i x_{t-i} - \sum_{j=1}^{q} \gamma_j \hat{x}_{t-j}.$$

We see that for all indices $i \leq \min(p, q)$ the common factor of $x_{t-i}$ is $\beta_i + \gamma_i$, if $p > q$ and $i > q$ the factor is $\beta_i$ and if $q > p$ and $i > p$ then the factor is $\gamma_i$, yielding equation 3.

## B   Further details and results for numerical experiments

### B.1   ARMA parameter recovery

In order to investigate if the implemented cell recovers parameters of an arbitrary ARMA model with coefficients estimated in a standard ARMA software (Seabold & Perktold (2010)), we simulate (V)ARMA processes for $25,000$ time steps and all possible combinations of $p, q \in \{0, 1, \ldots, 5\}$. We then train a neural network defined by a single linear ARMA cell on the data and check the convergence against the values obtained by maximum likelihood estimation. Results confirm that the ARMA cell can recover the coefficients for different values of $p$ and $q$, and also in the multivariate setting. Figure 4 visualizes one exemplary learning process.

### B.2   Elman parameter recovery

We now demonstrate the equivalence of a network based on the ARMA cell and an Elman network when only one MA lag is considered, i.e., when the ARMA cell is restricted to $q = 1$. For this, we take the ARMA(1,1) time series process used also in our simulation studies and fit linearly activated single-unit models based on both the ARMA cell and the Elman network. As shown in Figure 4, both models converge to the same parameters when applying the ARMA coefficient reparametrization as in equation 3.

### B.3   Description of simulated data generating processes

All error terms are a Gaussian white noise $\varepsilon_t \sim \mathcal{N}(0, 1)$. The data generating processes were defined as follows:

- ARMA(2,1)

$$x_t = 0.1x_{t-1} + 0.3x_{t-2} - 0.4\varepsilon_{t-1} + \varepsilon_t$$

- Threshold autoregressive (TAR)

$$x_t = \begin{cases} 0.9x_{t-1} + \varepsilon_t & \text{for } |x_{t-1}| \leq 1, \\ -0.3x_{t-1} + \varepsilon_t & \text{for } |x_{t-1}| > 1 \end{cases}$$

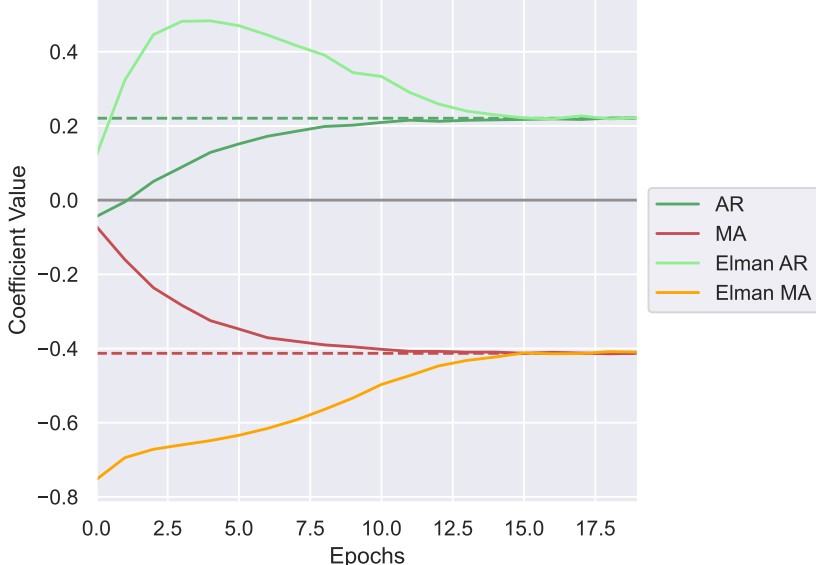

Figure 4: Examplary optimization paths for a single linear ARMA(1,1) and Elman cell using stochastic gradient descent. After around 30 iterations, the models converge to the maximum likelihood coefficients.

- Sign autoregressive (SGN)

$$x_t = sgn(x_{t-1}) + \varepsilon_t,$$

with

$$sgn(x) = \begin{cases} 1 \text{ for } x > 0, \\ 0 \text{ for } x = 0, \\ -1 \text{ for } x < 0 \end{cases}$$

- Non-linear autoregressive (NAR)

$$x_t = \frac{0.7|x_{t-1}|}{|x_{t-1} + 2|} + \varepsilon_t$$

- Heteroskedastic MA(2)

$$x_t = \varepsilon_t - 0.4\varepsilon_{t-1} + 0.3\varepsilon_{t-2} + 0.5\varepsilon_t\varepsilon_{t-2}$$

- VARMA

$$\begin{bmatrix} x_{t,1} \\ x_{t,2} \end{bmatrix} = \begin{bmatrix} 0.1 & -0.2 \\ -0.2 & 0.1 \end{bmatrix} \begin{bmatrix} x_{t-1,1} \\ x_{t-1,2} \end{bmatrix} + \begin{bmatrix} -0.4 & 0.2 \\ 0.2 & -0.4 \end{bmatrix} \begin{bmatrix} \varepsilon_{t-1,1} \\ \varepsilon_{t-1,2} \end{bmatrix} + \begin{bmatrix} \varepsilon_{t,1} \\ \varepsilon_{t,2} \end{bmatrix}$$

- Square multivariate (SQ)

$$x_{t,1} = 0.6x_{t-1} + \varepsilon_{t,1}$$
$$x_{t,2} = x_{t,1}^2 + \varepsilon_{t,2}$$

- Exponential multivariate (EXP)

$$x_{t,1} = 0.6x_{t-1} + \varepsilon_{t,1}$$
$$x_{t,2} = \exp(x_{t,1}) + \varepsilon_{t,2}$$

For the multivariate time series (VARMA, SQ, EXP), the second index of $x_{t,i}$, $i \in \{1, 2\}$, refers to the individual components.

## B.4 Ablation studies

### B.4.1 Time series length

| | ARMA 200 | ARMA 1k | ARMA 10k | TAR 200 | TAR 1k | TAR 10k | SGN 200 | SGN 1k | SGN 10k | NAR 200 | NAR 1k | NAR 10k | Hetero 200 | Hetero 1k | Hetero 10k |
|---|---|---|---|---|---|---|---|---|---|---|---|---|---|---|---|
| ARMA | 2.19±0.97 | 2.07±0.29 | 2.04±0.23 | 1.77±0.86 | 1.98±1.98 | 1.60±0.94 | 1.80±1.54 | 1.71±1.80 | 1.94±3.10 | 1.04±0.15 | 1.94±3.46 | 1.05±0.14 | 1.38±0.56 | 1.48±1.60 | 1.16±0.11 |
| ShallowARMA | **2.00±0.21** | **2.02±0.11** | **2.00±0.03** | **1.51±0.46** | **1.12±0.14** | **1.01±0.01** | **1.29±0.10** | 1.12±0.05 | 1.06±0.03 | *1.03±0.13* | **1.00±0.05** | **1.00±0.01** | **1.17±0.15** | **1.11±0.06** | 1.06±0.02 |
| DeepARMA | *2.04±0.22* | *2.04±0.11* | 2.00±0.04 | 1.71±0.43 | *1.18±0.29* | 1.06±0.23 | *1.32±0.16* | **1.07±0.05** | **1.02±0.03** | 1.03±0.13 | *1.00±0.05* | *1.00±0.01* | *1.21±0.17* | *1.12±0.06* | 1.05±0.02 |
| LSTM | 2.08±0.24 | 2.06±0.12 | *2.00±0.03* | 2.05±0.98 | 1.22±0.25 | 1.06±0.23 | 1.40±0.16 | 1.16±0.10 | *1.03±0.01* | 1.03±0.14 | 1.00±0.05 | 1.00±0.01 | 1.22±0.17 | 1.15±0.06 | 1.04±0.02 |
| DeepLSTM | 2.09±0.24 | 2.11±0.14 | 2.02±0.06 | 2.03±0.51 | 1.18±0.20 | 1.11±0.33 | 1.46±0.17 | 1.16±0.11 | 1.06±0.12 | 1.03±0.13 | 1.00±0.05 | 1.01±0.01 | 1.24±0.17 | 1.17±0.09 | 1.04±0.04 |
| GRU | 2.08±0.24 | 2.06±0.13 | 2.01±0.04 | *1.71±0.49* | 1.29±0.30 | 1.19±0.45 | 1.34±0.16 | 1.14±0.09 | 1.07±0.12 | 1.03±0.13 | 1.00±0.05 | 1.00±0.01 | 1.23±0.17 | 1.13±0.06 | *1.04±0.02* |
| DeepGRU | 2.08±0.22 | 2.08±0.13 | 2.01±0.04 | 1.77±0.43 | 1.21±0.26 | 1.19±0.44 | 1.38±0.15 | *1.11±0.09* | 1.04±0.10 | **1.03±0.13** | 1.00±0.05 | 1.00±0.01 | 1.23±0.18 | 1.13±0.07 | **1.03±0.02** |
| SIMPLE | 2.32±0.69 | 2.07±0.13 | 2.02±0.04 | 2.14±0.91 | 1.31±0.25 | *1.02±0.02* | 1.43±0.22 | 1.18±0.11 | 1.08±0.08 | 1.16±0.53 | 1.01±0.05 | 1.00±0.01 | 1.34±0.36 | 1.15±0.08 | 1.06±0.02 |
| DeepSIMPLE | 2.30±0.87 | 2.09±0.12 | 2.02±0.06 | 2.34±1.47 | 1.43±0.45 | 1.15±0.37 | 1.70±0.81 | 1.16±0.11 | 1.04±0.08 | 1.10±0.29 | 1.01±0.06 | 1.00±0.01 | 1.25±0.18 | 1.15±0.08 | 1.05±0.02 |

Table 7: Comparisons of different methods (rows) and different data generating processes (columns) across different time series lengths (200, 1000, 10000) for univariate time series using the average RMSE ± the standard deviation of 30 independent runs. The best performing method is highlighted in bold, the second-best in italics.

| | VARMA 200 | VARMA 1k | VARMA 10k | EXP 200 | EXP 1k | EXP 10k | SQ 200 | SQ 1k | SQ 10k |
|---|---|---|---|---|---|---|---|---|---|
| VARMA | 1.02±0.08 | **1.00±0.03** | **1.00±0.01** | 3.11±1.67 | 2.92±0.61 | 3.04±0.39 | 2.07±0.44 | 1.88±0.17 | 1.94±0.06 |
| ShallowARMA | **1.01±0.06** | *1.01±0.04* | *1.00±0.01* | **2.97±1.68** | **2.70±0.63** | **2.80±0.41** | **1.97±0.41** | **1.75±0.16** | *1.78±0.05* |
| DeepARMA | 1.02±0.07 | 1.01±0.04 | 1.00±0.01 | *3.02±1.68* | *2.71±0.63* | *2.81±0.41* | *2.04±0.47* | *1.75±0.15* | **1.78±0.05** |
| LSTM | *1.01±0.06* | 1.02±0.04 | 1.00±0.01 | 3.24±1.77 | 2.80±0.65 | 2.84±0.46 | 2.17±0.57 | 1.81±0.17 | 1.79±0.06 |
| DeepLSTM | 1.02±0.06 | 1.04±0.04 | 1.01±0.01 | 3.12±1.70 | 2.86±0.63 | 2.90±0.46 | 2.16±0.56 | 1.84±0.17 | 1.81±0.07 |
| GRU | 1.02±0.07 | 1.02±0.04 | 1.00±0.01 | 3.12±1.72 | 2.76±0.61 | 2.82±0.41 | 2.06±0.47 | 1.78±0.17 | 1.79±0.06 |
| DeepGRU | 1.02±0.06 | 1.02±0.04 | 1.00±0.01 | 3.15±1.73 | 2.81±0.66 | 2.87±0.43 | 2.14±0.55 | 1.81±0.18 | 1.80±0.06 |
| SIMPLE | 1.09±0.13 | 1.03±0.04 | 1.00±0.01 | 3.60±2.04 | 3.18±1.67 | 2.83±0.42 | 2.79±1.59 | 1.82±0.19 | 1.79±0.05 |
| DeepSIMPLE | 1.06±0.09 | 1.04±0.04 | 1.01±0.01 | 3.32±1.83 | 2.89±0.72 | 2.86±0.41 | 3.00±3.95 | 1.85±0.18 | 1.80±0.07 |

Table 8: Comparisons of different methods (rows) and different data generating processes (columns) across different time series lengths (200, 1000, 10000) for multivariate time series using the average RMSE ± the standard deviation of 30 independent runs. The best performing method is highlighted in bold, the second-best in italics.

| | ARMA 1 | ARMA 10 | ARMA 20 | TAR 1 | TAR 10 | TAR 20 | SGN 1 | SGN 10 | SGN 20 | NAR 1 | NAR 10 | NAR 20 | Hetero 1 | Hetero 10 | Hetero 20 |
|---|---|---|---|---|---|---|---|---|---|---|---|---|---|---|---|
| ARMA | 2.07±0.29 | **2.12±0.06** | **2.10±0.07** | 1.98±1.98 | 2.58±0.64 | 3.25±1.38 | 1.71±1.80 | 1.44±0.08 | 1.43±0.08 | 1.94±3.46 | 1.03±0.04 | 1.02±0.02 | 1.48±1.60 | 1.28±0.06 | 1.24±0.05 |
| ShallowARMA | **2.02±0.11** | 2.16±0.14 | 2.17±0.13 | **1.12±0.14** | 2.33±0.34 | 2.33±0.32 | 1.12±0.05 | 1.41±0.04 | **1.42±0.05** | **1.00±0.05** | 1.00±0.05 | 1.00±0.04 | **1.11±0.06** | 1.24±0.08 | 1.24±0.09 |
| DeepARMA | *2.04±0.11* | 2.17±0.13 | 2.17±0.13 | *1.18±0.29* | *2.26±0.33* | 2.32±0.33 | **1.07±0.05** | *1.41±0.04* | *1.42±0.05* | *1.00±0.05* | **1.00±0.05** | 1.00±0.04 | *1.12±0.06* | 1.24±0.08 | *1.23±0.09* |
| LSTM | 2.06±0.12 | 2.17±0.13 | 2.17±0.13 | 1.22±0.25 | 2.48±1.22 | 2.59±1.45 | 1.16±0.10 | 1.41±0.04 | 1.43±0.06 | 1.00±0.05 | 1.00±0.05 | 1.00±0.04 | 1.15±0.06 | 1.24±0.08 | 1.24±0.09 |
| DeepLSTM | 2.11±0.14 | 2.16±0.13 | *2.17±0.13* | 1.18±0.20 | 2.30±0.37 | 2.33±0.35 | 1.16±0.11 | **1.41±0.04** | 1.43±0.05 | 1.00±0.05 | 1.00±0.05 | **1.00±0.04** | 1.17±0.09 | *1.24±0.08* | **1.23±0.09** |
| GRU | 2.06±0.13 | 2.17±0.13 | 2.17±0.13 | 1.29±0.30 | **2.25±0.30** | **2.32±0.31** | 1.14±0.09 | 1.41±0.05 | 1.44±0.06 | 1.00±0.05 | 1.00±0.05 | 1.01±0.04 | 1.13±0.06 | 1.24±0.08 | 1.24±0.09 |
| DeepGRU | 2.08±0.13 | *2.16±0.13* | 2.17±0.13 | 1.21±0.26 | 2.29±0.35 | *2.32±0.32* | *1.11±0.09* | 1.41±0.04 | 1.43±0.06 | 1.00±0.05 | *1.00±0.05* | *1.00±0.04* | 1.13±0.07 | **1.24±0.08** | 1.24±0.09 |
| SIMPLE | 2.07±0.13 | 2.18±0.14 | 2.18±0.14 | 1.31±0.25 | 2.28±0.28 | 2.44±0.45 | 1.18±0.11 | 1.42±0.05 | 1.44±0.06 | 1.01±0.05 | 1.01±0.06 | 1.01±0.04 | 1.15±0.08 | 1.29±0.15 | 1.26±0.10 |
| DeepSIMPLE | 2.09±0.12 | 2.22±0.23 | 2.18±0.13 | 1.43±0.45 | 2.34±0.42 | 2.42±0.43 | 1.16±0.11 | 1.43±0.07 | 1.44±0.07 | 1.01±0.06 | 1.02±0.06 | 1.07±0.23 | 1.15±0.08 | 1.25±0.08 | 1.24±0.09 |

Table 9: Comparisons of different methods (rows) and different data generating processes (columns) across different forecasting horizons (1, 10, 20) for univariate time series using the average RMSE ± the standard deviation of 30 independent runs. The best performing method is highlighted in bold, the second-best in italics.

| | VARMA 1 | VARMA 10 | VARMA 20 | EXP 1 | EXP 10 | EXP 20 | SQ 1 | SQ 10 | SQ 20 |
|---|---|---|---|---|---|---|---|---|---|
| VARMA | **1.00±0.03** | **1.05±0.03** | **1.05±0.04** | 2.92±0.61 | **2.98±0.80** | **2.88±0.88** | 1.88±0.17 | **1.89±0.19** | **1.97±0.20** |
| ShallowARMA | *1.01±0.04* | 1.05±0.03 | 1.06±0.04 | **2.70±0.63** | 3.03±0.81 | *2.91±0.90* | **1.75±0.16** | *1.91±0.19* | *2.00±0.20* |
| DeepARMA | 1.01±0.04 | 1.05±0.04 | 1.05±0.04 | *2.71±0.63* | *3.02±0.81* | 2.91±0.89 | *1.75±0.15* | 1.91±0.20 | 2.00±0.20 |
| LSTM | 1.02±0.04 | 1.05±0.04 | 1.05±0.04 | 2.80±0.65 | 3.08±0.83 | 2.93±0.90 | 1.81±0.17 | 1.93±0.20 | 2.04±0.27 |
| DeepLSTM | 1.04±0.04 | *1.05±0.04* | *1.05±0.04* | 2.86±0.63 | 3.06±0.81 | 2.97±0.95 | 1.84±0.17 | 1.93±0.21 | 2.01±0.20 |
| GRU | 1.02±0.04 | 1.05±0.03 | 1.05±0.04 | 2.76±0.61 | 3.06±0.82 | 2.98±1.06 | 1.78±0.17 | 1.92±0.20 | 2.01±0.20 |
| DeepGRU | 1.02±0.04 | 1.05±0.04 | 1.05±0.04 | 2.81±0.66 | 3.06±0.83 | 2.93±0.88 | 1.81±0.18 | 1.91±0.19 | 2.01±0.21 |
| SIMPLE | 1.03±0.04 | 1.06±0.04 | 1.06±0.04 | 3.18±1.67 | 3.09±0.83 | 3.06±0.93 | 1.82±0.19 | 1.98±0.22 | 2.09±0.27 |
| DeepSIMPLE | 1.04±0.04 | 1.06±0.04 | 1.06±0.04 | 2.89±0.72 | 3.19±1.01 | 2.98±0.89 | 1.85±0.18 | 1.95±0.20 | 2.11±0.50 |

Table 10: Comparisons of different methods (rows) and different data generating processes (columns) across different forecasting horizons (1, 10, 20) for multivariate time series using the average RMSE ± the standard deviation of 30 independent runs. The best performing method is highlighted in bold, the second-best in italics.

## B.5 Description of benchmark datasets

**m4** Stemming from the Makridakis Competitions Makridakis et al. (2018) (see `https://en.wikipedia.org/wiki/Makridakis_Competitions` for more information), the m4 dataset contains 414 time series of hourly data. Every time series has a different starting point and a length of 748 hours. To allow for multivariate prediction, we take a subset of ten times series. overlap. We further take differences with a period of one and 24 hours to improve stationarity and reduce seasonal effects, respectively.

**Traffic** The traffic dataset can be downloaded from `https://archive.ics.uci.edu/ml/datasets/PEMS-SF`. It consists of 963 car lane occupancy rates with values between 0 and 1 taken from freeways in the San Francisco bay area. Time series start on the first of January 2008 and last until March 30 2009 with an observation frequency of 10 minutes. To condense the information, an hourly aggregation is used Yu et al. (2016), yielding time series of length 10,560. We use the first ten time series and observations until '2008-06-22 23:00:00', yielding a total of 4,167 observations per lane. We further apply seasonal differencing with a seasonal period of 24 hours and take first differences to reduce non-stationary behavior.

**Electricity** The electricity dataset can be downloaded from `https://archive.ics.uci.edu/ml/datasets/ElectricityLoadDiagrams20112014`. The dataset consists of electricity consumption (kWh) time series of 370 customers. Values correspond to electricity usage in a frequency of 15 minutes. In our benchmarks, we aggregate the values to hourly consumption (see also Yu et al. (2016) for justification of this approach). We use a subset of ten customers and a time range from '2014-01-01 00:00:00' to '2014-09-07 23:00:00', yielding a total of 6,000 observations per customer . We further apply seasonal differencing with a period of 24 hours to reduce seasonal effects and take the first differences for stationarity reasons.

**Exchange** The dataset was made available by Lai et al. (2018). The time series are exchange rates from 8 countries for days between January 1990 and May 2013. We calculate the returns of the exchange rates to receive a stationary time series.

**Moving MNIST** The dataset can be downloaded from `https://www.cs.toronto.edu/~nitish/unsupervised_video/`. It contains $10,000$ video sequences. Each sequence consists of 20 frames that show two digits moving with a random speed and direction. The digits move independently, but intersect from time to time and bounce off the edges of the frame. The resolution of the frames is $(64 \times 64)$ pixels and the monochrome light intensity is encoded as an 8-bit integer. We draw a random subset of $1,000$ sequences from this dataset for our experiments.

**Taxi New York City and Bike New York City** The datasets are available at `https://github.com/haoxingl/DSAN`. The goal is crowd flow prediction on a given spatial window in new york city. The Taxi-NYC dataset was originally taken from NYC-TLC (`https://www1.nyc.gov/site/tlc/index.page`) and the Bike-NYC dataset from Citi-Bike (`https://www.citibikenyc.com/`). Both datasets consist of 60 days of trip records. For every trip, the start and end location and time is included.

For all datasets except the Taxi New York City and Bike New York City we use the first 70% for training, and the remaining 30% for testing the model. For the neural network models, 30% of the training data is used for validation. The two New York datasets are already split into training (1920 hours), validation (576 hours), and testing (960 hours) data.

## B.6 Architectures and search space

For uni- and multivariate time series, all neural networks contain one to two RNN layers of the respective RNN cell, yielding the shallow and deep versions of the models, respectively. The cells in each layer contain one to five units with a rectified linear activation function. In the ShallowARMA model, one cell is activated linearly as shown in Figure 2, resembling a hybrid model. A final fully connected layer with linear activation and appropriate output shape is used to match the dimensions of the time series. The lag values $p$ and $q$ are chosen from the interval $[1, 4]$. The loss function of all models is the mean squared error function. For training, the Adam(Kingma & Ba, 2014) optimizer is used in combination with an early stopping callback

to prevent overfitting. For all other model properties, the default values are used. For tensor-variate time series, batch normalization layers are added between the RNN layers, and adaptive learning is added to improve convergence. Each layer contains 64 filters, so a 2D convolution is added to reduce the number of channels appropriately.

## C   Computational environment

All experiments and benchmarks were carried out on an internal cluster. Uni- and multivariate time series were trained on a server with 10 vCPUs, running on an Intel(R) Xeon(R) Gold 6148 CPU @ 2.40GHz physical CPU and 48Gb allocated memory. Tensor-variate time series were trained on a server with 16 vCPUS, running on an Intel(R) Xeon(R) Gold 6226R CPU @ 2.90GHz, 32Gb allocated memory, and a Nvidia GeForce RTX 2080 Ti (11Gb).

## D   Additional detail on experimental setup

We now give additional details about our experimental setup, clarifying the preprocessing steps as well as the training routines.

### D.1   General setup and optimization

Across all simulations and benchmarks, we use the Adam(Kingma & Ba, 2014) optimizer with a learning rate of 1e-3, and momentum parameters $\beta_1 = 0.9$ and $\beta_2 = 0.999$. We use a batch size of 32, early stopping with patience 10 iterations, and run the model for a maximum of 100 epochs.

### D.2   Competitor architectures

The following abbreviations are used for the methods we compare the ARMA cell against:

- LSTM/GRU/SIMPLE: A single LSTM/GRU/SIMPLE cell with ReLU activation function and a sigmoid recurrent activation function. The kernel, recurrent, and bias initializers are chosen to be uniform, orthogonal, and zero, respectively. The number of units is chosen via hyperparameter optimization in the range of one to five.

- DeepLSTM/DeepGRU/DeepSIMPLE: This setup stacks two of the cells in their non-deep counterparts, with the first layer set to return sequences.

### D.3   Input and out formats of RNNs

The input of the RNN cells are subsequences of the time series $\boldsymbol{X}$. For a sequence length $s$, the input shape for a non-ARMA cell RNN is given by $s \times k$. The input of the ARMA cell has an additional dimension for the lagged inputs, i.e., $s \times k \times p$.

The non-ARMA cell RNN with $u$ units returns an $s \times u$ time series. In order to allow the ARMA cell to be stacked, we have an additional dimension in the output. Specified by the keyword argument `return_lags`, we either return $s \times (d * u) \times 1$ if `False`, or $s \times (d * u) \times q$ if `True`. The factor $d$ ensures that a single unit ARMA cell returns the same number of dimensions as its input, allowing it to capture a classical VARMA model.

### D.4   Specific setups

#### D.4.1   Simulation study

For the simulated time series, no preprocessing was necessary, as the data generating processes are all stationary and also of the same magnitude.

### D.4.2 Benchmarks

For the real-world datasets, preprocessing was performed to improve the convergence of all models. In particular, we first apply differencing to remove trends in the time series. We define $\Delta_k$ to be the difference to the $k$'th lag of the time series. For M4, traffic, and electricity, we use $\Delta_1\Delta_{24}\boldsymbol{X}$. For the exchange dataset, we simply use the percentage returns of the exchange rates, as common practice in finance literature. We then standardize the resulting time series with their empirical mean and variance.

**Convolutional benchmarks**   For the benchmarks involving a convolutional layer, we use a batch size of 5 due to the higher memory requirements of the larger training data sizes.

### D.4.3 Integration with state-of-the-art forecasting frameworks

For the DeepAR benchmark, we use a larger dataset by looking at the first 100 columns of the M4, traffic, and electricity datasets. The exchange dataset was not extended, as it only consists of 8 columns in total.

**Architectures**   We use the following two types of DeepAR architectures:

- SINGLE: The SINGLE DeepAR model consists of one RNN block (either LSTM or ARMA) with $4(1 + log(d))$ units for the LSTM and one unit for the ARMA cell, a dropout rate of 0.2, and returns a sequence that is further processed by a fully-connected layer with $s(1 + log(d))$ number of units where $s \in \{1, 4\}$ is a hyperparameter and tanh activation. The resulting output is then fed into a Gaussian distribution layer that multiplies the number of input units by two to define both a mean and standard deviation for all output units by multiplying the inputs with a weight matrix of respective size. For the ARMA cell-based DeepAR model, we have the additional hyperparameters $p$ and $q$, which we optimize over the range from 1 to 4, again only considering $p = q$ for computational reasons.

- STACKED: The STACKED model uses the same architecture as SINGLE but combines two of the RNN cells specified by the SINGLE model.

## D.5   Investigation of parameter influence

In the following, we investigate the influence of the number of parameters on the performance and provide a summary of the resulting hyperparameter optimization results.

**Relation between number of parameters and performance**   In Figure 5 we compare the logarithmic RMSE values of different models to the (logarithmic) number of parameters in the case of a one-step forecasting horizon. Whereas performance values are very similar for all models for the linear time series (ARMA, TAR), the ShallowARMA model yields better results than the SIMPLE, GRU, and LSTM models while having fewer parameters compared to the latter two architectures. The DeepARMA model often yields a similar or larger number of parameters compared to the other deep architectures, while also yielding smaller RMSE values, in particular for non-linear time series (NAR, SGN).

A similar result can be observed for different forecasting horizons (Figure 6. When comparing only the two ARMA-variants (Figure 7, the improvement using a deep instead of a shallow ARMA becomes apparent, but – as expected – only on the non-linear data sets (TAR, SGN) and larger time series (1000, 10000).

We now further investigate the chosen number of lags $p$ and $q$ in the hyperparameter optimization routine. Figure 8 summarizes the result of this analysis by plotting the number of runs in which a given value for $p$ or $q$ was chosen. As we have the simplifying assumption of only considering $p = q$ in the ShallowARMA and DeepARMA models, only one bar is given for each model. We find that the classical ARMA model chooses the correct value of $p = 2$ in the majority of cases. We further find that the DeepARMA model tends to use a lower number of lags compared to the ShallowARMA model, indicating that the second layer facilitates more complex model spaces that are otherwise captured by a longer lag structure.

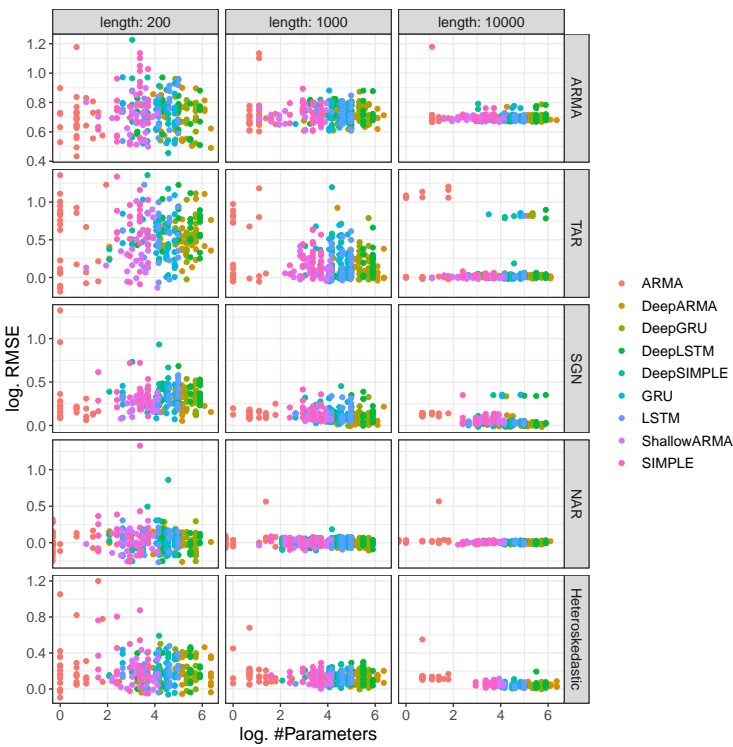

Figure 5: Logarithmic RMSE values (y-axis) for different models (colors) and their respective parameter numbers (x-axis) separated by the time series length (columns) and data sets (rows).

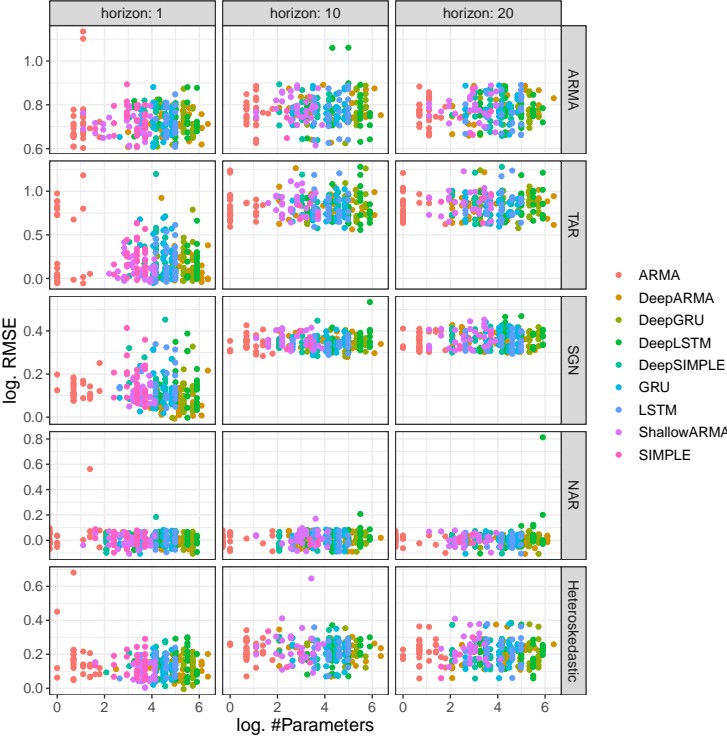

Figure 6: Logarithmic RMSE values (y-axis) for different models (colors) and their respective parameter numbers (x-axis) separated by the forecasting horizon (columns) and data sets (rows).

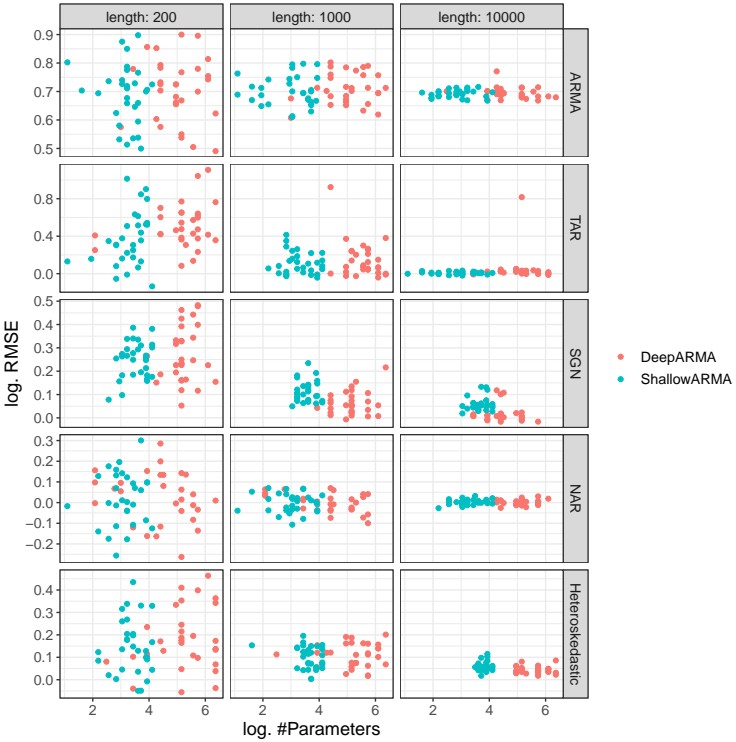

Figure 7: Logarithmic RMSE values (y-axis) for the two different ARMA models (colors) and their respective parameter numbers (x-axis) separated by the time series length (columns) and data sets (rows).

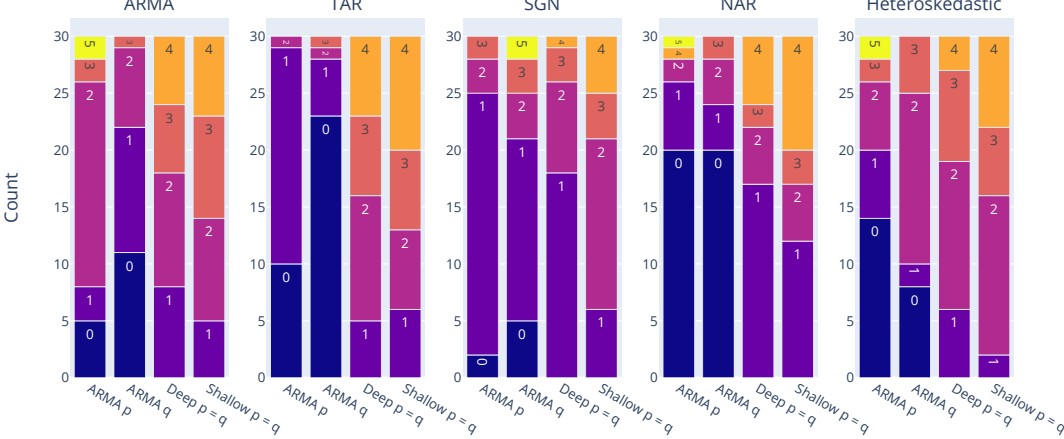

Figure 8: Aggregated lag choices (number of counts on the y-axis for $p$ and $q$ for the classical ARMA model, as well as the ShallowARMA and DeepARMA for each of the 5 data sets (different plots).

Table 11: Comparisons of different methods (rows) and different data generating processes (columns) for univariate time series using the average MAE ± the standard deviation of 10 independent runs. The best performing method is highlighted in bold, the second-best in italics.

| MODEL | ARMA | TAR | SGN | NAR | HETEROSKEDASTIC |
|---|---|---|---|---|---|
| ARMA | 1.62±0.29 | 2.39±2.66 | 1.98±2.35 | 2.28±4.42 | 0.95±0.11 |
| SHALLOWARMA | **1.55±0.07** | **0.84±0.06** | 0.88±0.06 | 0.81±0.04 | **0.88±0.05** |
| DEEPARMA | 1.57±0.08 | 0.96±0.38 | **0.84±0.05** | 0.81±0.04 | *0.88±0.05* |
| LSTM | 1.57±0.07 | 1.08±0.36 | 0.96±0.15 | *0.81±0.04* | 0.92±0.08 |
| DEEPLSTM | 1.60±0.08 | 1.12±0.42 | 0.94±0.12 | 0.81±0.04 | 0.92±0.07 |
| GRU | *1.56±0.09* | 0.98±0.22 | 0.88±0.07 | **0.81±0.04** | 0.90±0.06 |
| DEEPGRU | 1.58±0.07 | *0.95±0.26* | *0.87±0.10* | 0.81±0.04 | 0.90±0.05 |
| SIMPLE | 1.58±0.08 | 0.99±0.22 | 0.91±0.08 | 0.83±0.04 | 0.90±0.06 |
| DEEPSIMPLE | 1.60±0.08 | 1.15±0.46 | 0.92±0.08 | 0.82±0.04 | 0.92±0.08 |

**Influence of the non-linearity**   To assess the influence of the non-linear activation, we check how often a purely linear activation was chosen by the hyperparameter optimization for the ShallowARMA model in the univariate simulations. We find that the hyperparameter optimization opted to use non-linear activations in almost all cases, only using linear models 3 times for NAR, 1 time for Heteroskedastic, and 0 times for TAR and SGN. For the ARMA time series, however, a purely linear activation was selected more often, in 9 out of 30 cases. This indicates that the non-linearity contributes to the performance improvement of the cell. By contrast, we did not find substantial differences based on the number of units, as shown in Figures 5-7. Note that while these Figures show the number of parameters, this implicitly groups them by the number of units.

# E   Additional simulation and benchmark results

In the following, we provide additional results on numerical experiments by including comparisons based on the mean absolute error (MAE).

**Results**   The results for simulated data suggest that either the ShallowARMA or DeepARMA cell perform best in most cases while on par with the GRU cell for the ARMA and NAR dataset. For the simulated multivariate time series, none of the existing neural methods outperforms the ARMA cells. For the time series benchmark datasets, the ARMA approaches outperform all other RNN approaches on m4 and Traffic. On the Electricity dataset the ARMA cells remain competitive for the univariate case, but yield larger RMSE values compared to GRU and Simple in the multivariate setting. On Exchange all methods perform equally well.

Overall the rankings of methods do not change notably when using the MAE instead of the RMSE as comparison measure.

Table 12: Comparisons of different methods (rows) and different data generating processes (columns) for multivariate time series using the average MAE ± the standard deviation of 10 independent runs. The best performing method is highlighted in bold, the second-best in italics.

| MODEL | VARMA | EXP | SQ |
|---|---|---|---|
| VARMA | **0.80±0.02** | 1.61±0.13 | 1.22±0.07 |
| SHALLOWARMA | *0.80±0.03* | *1.48±0.13* | **1.19±0.06** |
| DEEPARMA | 0.81±0.03 | **1.48±0.13** | *1.20±0.06* |
| LSTM | 0.81±0.03 | 1.55±0.13 | 1.25±0.09 |
| DEEPLSTM | 0.82±0.03 | 1.63±0.16 | 1.29±0.10 |
| GRU | 0.81±0.03 | 1.53±0.14 | 1.23±0.07 |
| DEEPGRU | 0.81±0.03 | 1.54±0.13 | 1.23±0.09 |
| SIMPLE | 0.82±0.02 | 1.56±0.15 | 1.23±0.07 |
| DEEPSIMPLE | 0.82±0.03 | 1.56±0.17 | 1.25±0.08 |

Table 13: Comparison of different univariate and multivariate forecasting approaches (rows) for different datasets (columns) based on the average MAE ± the standard deviation of 10 independent runs. The best performing method is highlighted in bold, the second-best in italics.

| | | M4 | TRAFFIC | ELECTRICITY | EXCHANGE |
|---|---|---|---|---|---|
| UNIV. | ARMA | **0.82±0.00** | 0.48±0.00 | 0.77±0.00 | 0.78±0.00 |
| | SHALLOWARMA | *0.83±0.00* | 0.48±0.00 | 0.74±0.01 | 0.67±0.00 |
| | DEEPARMA | 0.84±0.01 | **0.45±0.01** | *0.70±0.02* | **0.67±0.00** |
| | LSTM | 0.88±0.03 | *0.45±0.00* | 0.72±0.03 | *0.67±0.00* |
| | DEEPLSTM | 0.98±0.10 | 0.45±0.00 | **0.69±0.02** | 0.67±0.00 |
| | GRU | 0.86±0.02 | 0.45±0.01 | 0.70±0.02 | 0.67±0.00 |
| | DEEPGRU | 0.86±0.02 | 0.45±0.01 | 0.70±0.01 | 0.67±0.00 |
| | SIMPLE | 0.91±0.06 | 0.46±0.00 | 0.71±0.01 | 0.67±0.00 |
| | DEEPSIMPLE | 0.93±0.07 | 0.46±0.01 | 0.70±0.01 | 0.67±0.00 |
| MULTIV. | ARMA | *0.82±0.00* | 0.48±0.00 | 0.77±0.00 | 0.78±0.00 |
| | SHALLOWARMA | **0.82±0.01** | 0.53±0.01 | 0.78±0.01 | 0.68±0.01 |
| | DEEPARMA | 0.83±0.00 | 0.53±0.02 | 0.77±0.03 | 0.68±0.00 |
| | LSTM | 0.97±0.04 | 0.49±0.01 | 0.93±0.20 | **0.67±0.00** |
| | DEEPLSTM | 1.06±0.08 | 0.49±0.02 | 0.74±0.03 | 0.69±0.06 |
| | GRU | 0.97±0.09 | 0.48±0.00 | 0.73±0.02 | 0.67±0.00 |
| | DEEPGRU | 0.97±0.06 | 0.48±0.01 | *0.72±0.02* | *0.67±0.00* |
| | SIMPLE | 0.99±0.03 | **0.47±0.01** | 0.73±0.02 | 0.67±0.00 |
| | DEEPSIMPLE | 1.01±0.05 | *0.47±0.01* | **0.70±0.01** | 0.67±0.00 |

