# OpenReview forum: "ARMA Cell: A Modular and Effective Approach for Neural Autoregressive Modeling"
_TMLR — Rejected by TMLR_

### Review · Reviewer_enze · 2022-09-28

**Summary Of Contributions:**

The paper proposes an RNN cell that implements an ARMA model, as a simpler alternative to LSTM or GRU cells for time series modeling applications. Additionally, a multivariate extension (ConvARMA) that replaces linear functions with convolutions is proposed, as well as networks that combine multiple such ARMA cells within a layer or through stacking. The empirical evaluation on synthetic time series as well as standard forecasting benchmark datasets demonstrates that the proposed approach can outperform other RNN cell types and classical ARMA models.

**Requested Changes:**

* [critical] Evaluation of the proposed cell in the context of a state-of-the-art RNN-based model and comparison to strong baselines. For example, take a DeepAR or LSTNet or even a Temporal Fusion Transformer model, and replace the RNN cells in that model with the proposed ARMA cell. How does the performance compare? Is it more stable? Requires fewer parameters? Is faster to train? More interpretable? This is currently listed as "future work" in the paper, but I would argue that without this comparison, the paper does not meet TMLR's "interestingness" criterion. Knowing that a tiny model made up of ARMA cells can outperform a tiny model made up of other kinds of RNN cells on some tiny datasets is by itself not particularly interesting. Alternatively, if replacing LSTM/GRU cells in such contexts is not the goal of the proposed ARMA cell, describe and evaluate real applications where using this kind of cell would be advantageous.
* [critical] Evaluate the effect of data set size (both observations per time series and number of time series) and model size (number of units, number of layers) on performance. Does the ARMA cell's performance advantage diminish with more data/larger models?
* [critical] Elucidate the relationship of the proposed cell to an Elman cell.
* [recommended] Clarify cell computations and $\hat{x}$ notation
* [recommended] Clarify experimental setup details: global vs. local model, forecast horizon, data normalization; number of parameters, training times;
* [recommended] Analyze and explain the performance difference between the classical ARMA model and the ShallowARMA. Is it due to the nonlinearity? Using multiple units? The optimization procedure?

**Strengths And Weaknesses:**

Strengths:
* Interpreting an ARMA model as an RNN cell that can be used as an alternative to other RNN cells is interesting and appears novel.
* The provided code ensures reproducibility and facilitates practical applications by making ARMA models available in a deep learning framework.

Weaknesses:
* Motivation and focus: The motivation behind the proposed ARMA cell is somewhat unclear. Being "simpler" than a LSTM or GRU cell does not by itself seem like a sufficient motivation. Showing that an ARMA model can be interpreted as an RNN cell and implemented and optimized within standard deep learning frameworks is interesting, but doesn't seem to be the main focus of the paper and is only explored superficially (e.g. no experiments are performed to explain the large performance gap between a "classical" ARMA model and the ARMA cell).
* Clarity: The writing is mostly clear and easy to follow. However, some details are missing or not stated clearly. For example, the operations performed by the proposed ARMA cell should be explicitly stated (i.e. how the output and updated state is computed in terms of the input and previous state). Further, the notation $\hat{x}$ is currently overloaded to mean both the random variable $x - \epsilon$ as well as the model predictions. In the figure it is also used to denote the output of a multi-unit or multi-layer network, which makes it confusing to understand what is actually used as the recurrent input to the cell. Initialization of the cell and its effect is not discussed.
* Relationships to other RNN cells not described: The relationship of the proposed ARMA cell to other RNN types is alluded to, but not explicitly stated. Unless I am misunderstanding something, the ARMA cell seems to be a special case of an Elman RNN where the weight matrices have a special restricted form. If this is the case, this connection should be explicitly described; if not, the differences should be highlighted.
* Empirical evaluation has several shortcomings:
  - Some details of the evaluation setup are not clearly described, e.g. global vs. local models, forecast horizon, data normalization.
  - The synthetic experiments seem to be performed in a setting where the models are trained on a single fixed-length (1000 steps) time series. This makes it difficult to understand the bias-variance trade-off made by the ARMA cell. Intuively, I would expect the simpler ARMA cell to perform better when only small amounts of training data are available (due to smaller variance), while the performance of the more complex RNN cells should improve (and potentially ultimately outperform) the ARMA cell as more training data becomes available. Would be very interesting to see this explored in an experiment where the training set size is varied.
  - The models considered in the experiments seem to be extremely small (1-2 layers, 1-5 units); how does the performance change if larger models are considered? A related question: how does the number of free parameters differ between the different cells? Knowing the relative performance difference between two kinds of small models is by itself not particularly interesting if state-of-the-art performance is typically only achieved with much larger models.
  - The evaluation on the real data sets seems to be performed in a "local" fashion, where a model is fit for each time series separately. This setup tends to favor simpler models, while deep models typically shine when trained globally, i.e. a single univariate model is trained across all time series in the data set. Current deep learning forecasting models are almost exclusively trained as global models.
  - The proposed cell is not evaluated in the context of a state-of-the-art model or compared to strong baselines. The authors argue this is by design, as the complexity of such models would preclude one from making "statement[s] about the performance of a single recurrent cell therein". I would argue that evaluating the performance of a single cell is not meaningful (as noone uses such simple models in practice, and it is not clear that single cell performance carries over to the performance of larger networks), and what one cares about is the performance of a model composed of such cells on a real-world task.

---

> ### Author Response · Authors · 2022-11-24
> **Response and revision (part 1)**
>
> We thank the reviewer for taking the time to carefully read our paper and for pointing out ways to improve our manuscript.
>
> ## Summary
> - **[Motivation]**: We have added an additional paragraph, elaborating on the motivation of the ARMA cell (see below).
> - **[Clarity]**: We thank the reviewer for pointing out missing details and the overloaded $\hat{x}$ notation, which we have now added to the manuscript.
> - **[Relationship]**: We now highlight the relationship to existing approaches more clearly (see also our answer to your requested changes below).
> - **[Empirical evaluation]**: As suggested in your requested changes section, we have added various experiments to our paper (see below). We would also like to point out the additional results that have been added to the paper in response to requested changes from reviewer *hpio*.
> All in all, these changes have improved the paper significantly and made the paper much more complete.
>
> ## Requested changes
> We respond to each requested change by enumerating them in the given order from 1-6.
>
> ### Request 1: State-of-the-art comparison
> We appreciate this comment and agree that adding a comparison to a state-of-the-art framework would improve the paper significantly, in particular by providing evidence for the modularity aspect of the paper.
> We have therefore added the subsection “Integration with state-of-the-art forecasting frameworks” to showcase integration with an implementation of the DeepAR framework. As this makes a fair comparison even more challenging, we focused on the integration alongside first evidence of its efficacy by comparing an LSTM- and an ARMA-type DeepAR model for two different fixed model sizes (small and large) and tuning their hyperparameters using grid-search (in the same way as we did for our simulation study; details are given in our appendix).
> Again, the goal is not to show superior predictions but that the ARMA cell is able to perform equally well despite its more simple architecture. Below is an excerpt of the results:
>
> |                    | m4                          | traffic                     | electricity                 | exchange                    |
> |---------------------|-----------------------------|-----------------------------|-----------------------------|-----------------------------|
> | DeepAR ARMA 1-Layer | 2.444 $\pm$ 0.137           | **1.323 $\pm$ 0.033** | **5.236 $\pm$ 0.398** | 1.552 $\pm$ 0.025           |
> | DeepAR LSTM 1-Layer | **2.306 $\pm$ 0.056** | 1.362 $\pm$ 0.054           | 10.236 $\pm$ 9.242          | **1.510 $\pm$ 0.034** |
> | DeepAR ARMA 2-Layer | 2.513 $\pm$ 0.157           | **1.332 $\pm$ 0.056** | **5.639 $\pm$ 0.461** | 1.551 $\pm$ 0.026           |
> | DeepAR LSTM 2-Layer | **2.266 $\pm$ 0.049** | 1.381 $\pm$ 0.053           | 9.607 $\pm$  5.630          | **1.494 $\pm$ 0.033** |
>
>
> ### Request 2: Effects of dataset and model size
> We have added additional experiments to investigate the mentioned relationships. In particular, we observe that – as expected – the advantage of the ARMA cell over more complex neural cells is reduced for very long time series. For shorter time series it still fares well compared to the linear ARMA model in most cases.
> For the number of time series, we refer to our comparison of training univariate time series compared to a single multivariate model in section 5.3.1, suggesting that the additional hyperparameter tuning for each time series outweighed the additional information from training all time series in a single model. This can of course change if there are stronger dependencies between the individual series.
> We have also added an ablation study to investigate how model complexity affects forecasting performance in section 5.2 (new), finding that
> > “the rank of the different methods is similar to the aforementioned results, and ShallowARMA yields the best results in most settings. There is, however, a clear trend in that the performance differences between the different cell types become irrelevant for an increase in $T$.”
>
> We further find that
>
> > “for forecasting horizons greater one, the different ARMA variations do not outperform other approaches anymore and DeepLSTM, GRU, or DeepGRU yield the best results in many cases. The performance values, however, are in most cases within one standard deviation of those by the Shallow- or DeepARMA approach. For the multivariate case, the classical VARMA model provides the best forecast for all multi-step ahead forecast scenarios, closely followed by the Shallow- and DeepARMA models.”

---

> ### Author Response · Authors · 2022-11-24
> **Response and revision (part 2)**
>
> ### Request 3: Relation to Elman cell
>
> We have added a paragraph for the comparisons between these cells as suggested by the reviewer.
>
> > “When considering a feedforward neural network with a single hidden layer, such cyclical connections can be established by concatenating the output of the previous time step to the inputs of the next, yielding an Elman network (Elman, 1990). The proposed ARMA cell includes the Elman network as a special case when setting the moving average part of the cell to the value 1. We also demonstrate this in our numerical experiments, showing that both models result in the same parameter estimates. Similarly, a Jordan network is obtained by concatenating the previous hidden layer to the subsequent input (Jordan, 1986).”
>
> ### Request 4: Notation
> We thank the reviewer for pointing this out. Section 4.1 was intended to describe the model, not the data-generating process.  Hence the definition of $\hat{x}$ as given in this section was changed to $\hat{x}_t = x_t - \hat{\varepsilon}_t$ to clarify that the model is working on the observed residual term $\hat{\varepsilon}_t$ and not the random variable $\varepsilon_t$.
>
> ### Request 5: Experimental setup
> We have now clarified the distinction between local and global models in our benchmarks:
>
> > For univariate time series, this is done by training a **local** model for every dimension and averaging the results over the different multivariate dimensions. The multivariate comparison is based on the predictions of a single **global** model.
>
> and
>
> > We then train a larger **global** model on the previously studied benchmark data sets for multivariate time series [...]
> We further describe our experimental setup in more detail in Section D and, in particular, investigate the effect of parameter sizes after hyperparameter tuning on the performance in Appendix D.5.
>
> ### Request 6: Performance difference to ARMA
> We have added this information to Appendix D.5, stating:
> > “We find that the hyperparameter optimization opted to use nonlinear activations in almost all cases, only using linear models 3 times for NAR, 1 time for Heteroskedastic, and 0 times for TAR and SGN. For the ARMA time series, however, a purely linear activation was selected more often, in 9 out of 30 cases. This indicates that the nonlinearity contributes to the performance improvement of the cell. By contrast, we did not find substantial differences based on the number of units, as shown in Figures 5-7. Note that while these Figures show the number of parameters, this implicitly groups them by the number of units.”

---

### Review · Reviewer_hpio · 2022-09-30

**Summary Of Contributions:**

The paper proposes an ARMA cell which is based on the classical autoregressive moving average models for time series modeling. This cell can be used in place of LSTM/GRU cell in recurrent neural architectures. The paper also introduces convolution based ARMA cell for spatially-correlated time series. Experiments on multiple datasets show that the ARMA cell based recurrent networks are very effective for modeling time series.

**Requested Changes:**

The paper should address the first five concerns mentioned in the weaknesses section for securing my acceptance recommendation.
Other concerns should be addressed to further strengthen the paper.

**Strengths And Weaknesses:**

### Strengths
- The paper presents an interesting idea with the use of classic ARMA model to introduce ARMA cell.
- Experiments show that the framework is able to outperform baselines on multiple datasets.
- The paper is easy to follow.

### Weaknesses
1. One of the important point that the paper tries to make is that the proposed model is simpler than standard recurrent cells. But from equation (3), it seems like it is very much similar to a standard RNN cell (in Elman/Jordan networks) albeit more complex when p or q is greater than 1. It becomes a standard RNN when p and q are equal to 1. Is the main difference between ARMA cell based RNN and simple RNN is the number of lags? The paper should perform ablation with p=1 and q=1.
2. I am sympathetic to the idea of fixing certain architectural choices, e.g., fixing certain hyperparameters to default because it (a) gives the appearance of a "fair comparison" and (b) reduces burden of effort, but I do not agree that it yields a truly fair comparison. It is possible that some of the baseline approaches could be overfitting/underfitting to the datasets. A truly fair comparison requires independently tuning all the hyperparameters for each model.
3. RNN based approaches are also application for long term forecasting. How does the proposed model perform in that case? The paper should provide more experiments to show when to use LSTM or GRU kind of approach and when to use ARMA cell.
4. Since one of the claims of the paper is that the proposed approach is modular, I would expect the paper to include ARMA cell in state-of-the-art forecasting frameworks to show its usefulness.
5. The training/evaluation protocol is not clear. Is it one-step forecasting or multi-step? What's the input and output to the RNN models?
6. It is not clear what values of p and q were selected for the experiments.
7. There are few cases where classic ARMA model outperforms the ARMA-cell based RNNs. Can the paper discuss in detail in what cases is this possible?
8. What is the baseline in Table 4?

---

> ### Author Response · Authors · 2022-11-24
> **Response and revision (part 1)**
>
> ## Requested changes
>
> We respond to each requested change by following the enumeration from 1-8.
>
> ### Request 1: Comparison to Simple RNN
> This is, indeed, a very relevant question. As noted correctly by the reviewer, the ARMA(1,1) cell is equivalent to the SimpleRNN cell, which in turn provides a generalization of the Elman network for q=1. The computations in the ARMA cell are, however, more complex when using autoregressive lags or q>1, yet with a simpler mechanism compared to GRU or LSTM cells. In addition, the ARMA cell is more in line with the usual time series model approaches and can replicate classical models if the cell is not stacked and only activated linearly. To make this more clear, we now provide a more detailed comparison between the ARMA cell, the simple RNN and Jordan/Elman networks in Section 2, and have added further comparisons between these cells as suggested by the reviewer.
>
> > “When considering a feedforward neural network with a single hidden layer, such cyclical connections can be established by concatenating the output of the previous time step to the inputs of the next, yielding an Elman network (Elman, 1990). The proposed ARMA cell includes the Elman network as a special case when setting the moving average part of the cell to the value 1. We also demonstrate this in our numerical experiments, showing that both models result in the same parameter estimates. Similarly, a Jordan network is obtained by concatenating the previous hidden layer to the subsequent input (Jordan, 1986).”
>
> ### Request 2: Hyperparameter optimization
> We agree with the reviewer that a truly fair comparison does not fix any hyperparameter settings. We, however, would like to emphasize that we have tuned most of the hyperparameters (e.g., number of units) either using exhaustive grid search or by making them explicit through the reported complexity of the model (i.e., instead of tuning the number of layers, we report models with different complexities). The hyperparameters that we have not considered in our experiments were only related to the optimization (i.e., type of optimizer, learning rate, or batch size). For these settings, we chose defaults that are currently used in practice while checking for all runs whether these optimization routines do actually converge. Given the ballpark of the different methods in our experiments, we would assume that even if tuned more extensively, the optimization will not change results notably and that the ARMA cell would still provide a competitive approach to GRU, LSTM and SimpleRNN in the analyzed settings. In order to make our tuning as transparent as possible (so that fairness can also be assessed by the readers of our work), we provide all codes for numerical experiments as well as all details on configurations in our comparisons in the Supplementary Material.
>
> To check whether methods over- or underfit to the data, we have now included further ablation studies (Section 5.2) that investigate the performances for different data sizes. These experiments suggest that even for larger or smaller data sets, a similar behavior as presented in the main table in our previous paper version can be observed.
>
> ### Request 3: Long-term forecasting
> We thank the reviewer for pointing this out. This is a relevant point and we have now extended our simulation experiments to multi-step forecasting. In fact, results suggest that the ARMA cell does not yield superior results for long-term forecasting compared to the LSTM or GRU cell, but does also not underperform in this case. We would therefore recommend the use of our ARMA cell in cases of short-term forecasting, but practitioners could also consider its use for long-term forecasting.

---

> ### Author Response · Authors · 2022-11-24
> **Response and revision (part 2)**
>
> ### Request 4: State-of-the-art comparison
> We appreciate this comment and agree that adding a comparison to a state-of-the-art framework would improve the paper significantly, in particular by providing evidence for the modularity aspect of the paper.
> We have therefore added the subsection “Integration with state-of-the-art forecasting frameworks” to showcase integration with an implementation of the DeepAR framework. As this makes a fair comparison even more challenging, we focused on the integration alongside first evidence of its efficacy by comparing an LSTM- and an ARMA-type DeepAR model for two different fixed model sizes (small and large) and tuning their hyperparameters using grid-search (in the same way as we did for our simulation study; details are given in our appendix).
>
> Again, the goal is not to show superior predictions but that the ARMA cell is able to perform equally well despite its more simple architecture. Below an excerpt of the results:
>
> |                    | m4                          | traffic                     | electricity                 | exchange                    |
> |---------------------|-----------------------------|-----------------------------|-----------------------------|-----------------------------|
> | DeepAR ARMA 1-Layer | 2.444 $\pm$ 0.137           | **1.323 $\pm$ 0.033** | **5.236 $\pm$ 0.398** | 1.552 $\pm$ 0.025           |
> | DeepAR LSTM 1-Layer | **2.306 $\pm$ 0.056** | 1.362 $\pm$ 0.054           | 10.236 $\pm$ 9.242          | **1.510 $\pm$ 0.034** |
> | DeepAR ARMA 2-Layer | 2.513 $\pm$ 0.157           | **1.332 $\pm$ 0.056** | **5.639 $\pm$ 0.461** | 1.551 $\pm$ 0.026           |
> | DeepAR LSTM 2-Layer | **2.266 $\pm$ 0.049** | 1.381 $\pm$ 0.053           | 9.607 $\pm$  5.630          | **1.494 $\pm$ 0.033** |
>
> ### Request 5: Training/evaluation protocol
> - We now provide the necessary information on the training and evaluation protocol in Appendix D.
> - In the current version of the paper we further explicitly distinguish between one-step and multi-step forecasting, first investigating one-step ahead forecasts and then analyzing the difference in results when using multi-step ahead prediction in Section 5.2.
> - We further explain the input of the RNN models in more detail in the Supplementary Material D.3, writing:
>
> > The input of the RNNs are subsequences of the time series $\boldsymbol{X}$. For a sequence length $s$, the input shape for a non-ARMA cell RNN is given by $s \times k$. The input of the ARMA cell has an additional dimension for the lagged inputs, i.e., $s \times k \times p$.
> The non-ARMA cell RNN with $u$ units returns an $s \times u$ time series. In order to allow the ARMA cell to be stacked, we have an additional dimension in the output. Specified by the keyword argument `return_lags`, we either return $s \times (d \cdot u) \times 1$ if `False`, or $s \times (d\cdot u) \times q$ if `True`. The factor $d$ ensures that a single unit ARMA cell returns the same number of dimensions as its input, allowing it to capture a classical VARMA model.
>
> ### Request 6: Selection of $p$ and $q$
> For all experiments, p and q were hyperparameters chosen between 1 and 4. We have now exported the tuned choices for p and q and analyzed them in Appendix D.5.
>
> ### Request 7: Performance comparison to ARMA
> We now discuss in detail in what cases this is possible in Section 5:
> > “In the case where the data generating process is in fact a (V)ARMA process, we expect the classical (V)ARMA model and the ARMA cell to perform similarly, but note that the optimization using stochastic gradient descent can sometimes yield better estimations of this process and hence outperform these classical models despite having the exact same hypothesis space.”
> In our experiments, the stochastic optimization yields inferior results in very few cases, which are however not significantly worse than the ARMA model and can be related to the stochasticity of the optimization routine.
>
> ### Request 8: Baseline ConvARMA
> Thank you for pointing this out. Indeed, we missed clarifying what the baseline in this case is. Since often in image sequences only a few pixels change, a reasonable baseline is to use the previous image as the prediction, which is in this case the baseline we compare against. We have added this to the paper
> > “As a baseline model, we report the performance of simply predicting the image of the previous frame.”
>
> # Summary
> In summary, the requested changes have notably enhanced the paper and we would like to thank the reviewer again for all the thoughtful comments.

---

### Review · Reviewer_Yqjy · 2022-11-21

**Summary Of Contributions:**

The paper proposes a new RNN for modelling time series data, based on a simplified cell structure based on ARMA models, as well as extrension for spatially-correlated time-series data.

**Broader Impact Concerns:**

No ethical concerns here.

**Requested Changes:**

Crucially, I think that the authors would need to strongly clarify the motivations of the ARMA cell in concrete terms, in particular:

1.  What kinds of representations can the ARMA cell learn that cannot be captured by ARMA modelling? If the main difference is the non-linear output, how is this different/better from using a link function instead?
2. Are there any types of relationships that the ARMA cells can capture that standard RNNs would have difficulty in learning?
3. Flesh out the advantages at the start of Section 4. Why is it important to have the same coefficients as classical ARMA? Why is ARMA computationally unfeasible on large datasets despite its simple structure? Which hybrid models cannot be trained end-to-end, and are they being compared to in the results?
3. Apart from better representations (of which evidence is insufficient), are there any theoretical results from ARMA that can be used for RNNs that would be beneficial?


In addition, the paper would require additional details on training, in particular how past predictions (i.e. x_hat) are fed into the model. Is a teacher-forcing approach taken?

**Strengths And Weaknesses:**

The ARMA process is extremely well-studied in the econometrics and signal processing literature, and any unifications of theoretical results with modern RNNs would have been very interesting -- particularly from a model interpretability stand point.

However, the motivations of modelling a ARMA process as an RNN cell are still not immediately apparent to me. In the summary, the authors do mention that sometimes complex RNNs under-perform, but 1) the improvements obtained, where present, are not statistically significant, and 2) given the simplicity of the basic ARMA cell why we should not simply use a standard ARMA model in its place.

---

> ### Author Response · Authors · 2022-11-24
> **Response and revision (part 1)**
>
> ## Requested changes
>
> We thank the reviewer for carefully reading our manuscript and the insightful comments as well as suggestions to improve our manuscript. We will address each of the four points in the following.
>
> ### Request 1: Learnable representations
> We would like to thank the reviewer for this question, as it was not explicitly mentioned in our paper before.
> Indeed, a linear ARMA cell shares the same hypothesis space as a classical ARMA model. It is also correct that for a single unit ARMA cell, the nonlinear outputs can be included in classical ARMA models via a link function.
> As mentioned in Section 4.2, the ARMA cells can be stacked both in a multi-unit way within a single layer, as well as by combining multiple layers consecutively. This allows the ARMA cell to learn representations that can not be captured by a single-layer and single-unit ARMA cell, or by a classical ARMA model, for that matter.
> We draw a parallel here to regular multi-layer perceptrons, where each node is a simple regression model with an activation function, which on their own can be modeled without the need for a deep learning framework. However, the combination of multiple such units increases the hypothesis space and makes the models more expressive.
> We have highlighted this distinction in Section 4.1:
> > “The above-mentioned ARMA cell has the same hypothesis space as the classical ARMA model when using a single-unit ARMA cell. While using a nonlinear activation for the outputs, in this case, is equivalent to using a link function (as done in generalized linear models) for the classical ARMA model, extensions using multiple units or stacking ARMA cells (see below) increase the model's expressiveness. As for regular multi-layer perceptrons, where each node is a simple regression model with an activation function and the combination of multiple such units makes the models more expressive, these extensions combine simpler ARMA models and therefore allow modeling more complex relationships.”
>
> ### Request 2: Comparison to RNNs
> This is certainly an important point to highlight. We have now included this comparison in more detail in Section~4.1, writing:
> > “In contrast to the standard RNN cell, the ARMA cell internally can access multiple previous states and lagged features, making it potentially easier to learn time dependencies and recurrences. The standard RNN cell, in contrast, only relies on the current input and the previous cell state. In other words, the ARMA cell allows for a richer autoregression and, in contrast to the simple RNN, provides a way to model moving averages. This can also be explained using Figure~1, where the standard RNN cell can represent the black arrows, but not the red and green connections.”

---

> ### Author Response · Authors · 2022-11-24
> **Response and revision (part 2)**
>
> ### Request 3: Advantages
>
> We have now extended the advantages of the ARMA cell in Section 4 by introducing a new paragraph and adding further details to the previously mentioned points in the original submission.
> - **[Importance of same coefficients and hybrid models]**: One motivation for the ARMA cell was to provide an RNN cell that – in its simplest form – provides a well established and principled baseline model (i.e. an ARMA model). This gives researchers a fallback property in case there is no non-linearity or higher complexity in the data. As most researcher use an automated hyperparameter optimization method, this fallback would be automatically selected, not only yielding a solid baseline instead of a possibly diverging or too complex model, but also provide further insights as parameters in the simple ARMA cell (which resembles the ARMA model) can be much better understood. Another motivation is simply convenience: Both the ARMA model and any other complex RNN can be modeled in one framework, allowing simpler benchmarks and fostering comparability by only needing to rely on one software framework. A third motivation is the possibility to straightforwardly combine models (e.g., a multilayer perceptron and an ARMA model) in one holistic network instead of performing a two-stage (and usually two-software) approach, which is more error-prone and usually does not jointly optimize the parameter of both model parts as it is done sequentially. As we have not provided an example for the latter motivation so far, we have now included another experiment in our paper, comparing the aforementioned two-step approach on all simulated data sets with a joint modeling approach. An excerpt of the results below, showing that in particular on the non-linear data set (NAR) the end-to-end approach outperforms the hybrid one. We thank the reviewer for pointing this out.
>
> |         | ARMA              | TAR               | SGN               | NAR               | Heteroskedastic   |
> |---------|-------------------|-------------------|-------------------|-------------------|-------------------|
> | End2End | 2.021 $\pm$ 0.105 | **1.134 $\pm$ 0.127** | **1.128 $\pm$ 0.055** | **1.002 $\pm$ 0.044** | **1.142 $\pm$ 0.062** |
> | Hybrid  | **2.014 $\pm$ 0.112** | 1.264 $\pm$ 0.811 | 1.138 $\pm$ 0.051 | 3.009 $\pm$ 8.869 | 1.147 $\pm$ 0.062 |
>
> - **[Computational feasibility for ARMA]**: The reviewer is correct in saying that the ARMA has a simple structure and should thus work also for large data sets. In fact, its complexity is only linear in n. Yet, for larger datasets, computations involved in the ARMA model might not be feasible on e.g. personal laptops or computers (both because statistical software usually works in memory and because models also scale more than quadratic with p and q). The ARMA cell does not have this problem as it optimizes the model in mini-batches, i.e., can work for arbitrarily large data sets. As time series are often rather short, mini-batch training however gets particularly beneficial when modeling multivariate time series (e.g., predicting electricity consumption of 100s of households as in the provided benchmark experiment) where the amount of data and the model costs (compared **V**ARMA model) grow much faster.\
> A second aspect that makes an ARMA cell potentially superior to an ARMA model in terms of computational feasibility is its robustness. This is clearly demonstrated by all our experiments, where the ARMA cell yields often better estimates and does this without failing in any of the runs, whereas the non-stochastic optimization (classical ARMA implementation) is less robust to ill-conditioning as well as numerical instabilities and failed to converge in numerous of our experiments.
>
> We highlighted these two advantages more clearly and thank the reviewer for bringing up this point.

---

> ### Author Response · Authors · 2022-11-24
> **Response and revision (part 3)**
>
> ### Request 4: Potential benefits
>
> While we can adopt statistical inference results (i.e., epistemic uncertainty quantification) from the time series literature for a simple 1-unit ARMA cell, this becomes more challenging for larger models. For larger models, a similar approach as presented in Immer et al., (2021; https://proceedings.mlr.press/v130/immer21a.html) could be used to perform last-layer inference in a stacked RNN model where the last cell is an ARMA cell. Although this neglects the variance in previous layers, it can provide a good indication of the network’s uncertainty. Further, when merging multiple linearly activated ARMA cells, the combination is an ensemble of ARMA models, for which some form of uncertainty quantification method could be derived (being a model average of additive models). These are only a few of possibly many links that could be used to transfer theoretical results from statistical time series analysis to RNNs. Due to the wealth of possible synergies, we have added a short outlook to our discussion to point out various of these interesting future research directions and thank the reviewer for making us aware of this particular aspect.
>
> > “As noted by an anonymous reviewer, an interesting future research direction is to make use of the theoretical results for ARMA models known from classical statistical literature and transfer these to the application of ARMA as a cell with multiple units or in its stacked variant. A directly available result, e.g., would be last-layer uncertainty quantification (see, e.g., Immer et al., 2021) in a stacked RNN model where the last cell is an ARMA cell with one unit. Although this neglects the variance in previous layers, it allows a first assessment of the RNN's uncertainty. Further, when merging multiple linearly activated ARMA cells, the combination is an ensemble of ARMA models, for which some form of uncertainty quantification method could be derived.”
>
> ## Additional comments:
> We agree with the reviewer that the paper requires additional details on training. We now have added another subsection to section 4 (before Extensions), describing our training procedure. In particular,
> - For a given sequence, the ARMA cell creates predictions by recursively applying formula (3). This is done in one forward pass. To also allow predictions for the first q time points in each sequence, we need to pad the sequence of previous predictions with 0-values.
> - We then differentiate the loss of these outputs given the current weights back through the whole sequence, i.e., the network is trained exactly as done for the LSTM, GRU and simple RNN cell via backpropagation through time. Teacher-forcing in our case would be possible, but potentially make it more difficult to learn the MA component by overwriting the predicted values with training data.
> - Note that our ARMA cell also supports returning sequences, which we can use to stack cells or for training a model on multiple steps simultaneously.
>
> # Summary
> In summary, we think that also the third review allowed us to further refine our manuscript and improve its significance by adding further discussion and results. We would like to thank the reviewer and hope that we have addressed all the mentioned comments satisfactorily.

---

### Author Response · Authors · 2022-11-24
**Summary of updates**

Dear AC and all Reviewers,\
We sincerely appreciate the AC’s and all reviewers’ time and insightful comments, which helped a lot in further improving our paper.

## Summary
We would like to summarize our revision and point out the improvements to our manuscript:
- **[Further Experiments and Analysis]**: We have added various additional experiments including 1) an ablation study to investigate a) the influence of the length of the time series and b) the influence of the forecasting horizon of the time series; 2) a comparison between an end-to-end hybrid model using the ARMA cell and a two-stage modeling approach; 3) a benchmark comparing the efficacy of the ARMA cell in the state-of-the-art forecasting architecture DeepAR. We have further analyzed the resulting findings more thoroughly by relating the number of parameters and the performance of models, and discovered what hyperparameters mostly drive the ARMA cells by looking into the frequency of hyperparameter choices in our experiments.
- **[Related Methods]**: We now provide more detailed explanations on how our method relates to Jordan/Elman networks, the standard RNN cell as well as the classical ARMA model.
- **[Clarity]**: We fixed some notational inconsistencies pointed out by the reviewers and added some key terms to make it easier for readers to understand the idea behind our method.
- **[Motivation]**: We improved the motivation for our proposed method, by highlighting its direct link to classical time-series forecasting, thereby providing a strong baseline for neural methods. Additionally, we point out the improved robustness and computational aspects arising from the ARMA cell.

Again, we greatly appreciate the time and efforts of reviewers and the AC and thank everyone for helping to improve our manuscript! We firmly believe that the ARMA cell is a promising candidate for improving time-series forecasting, and narrowing the gap between classical and neural methods.

Authors

---

### Decision · Action_Editors · 2023-02-09

**Recommendation:** Reject

**Comment:**

I apologize for the delay in this final decision.

While I agree with the reviewers that the premise of the paper is interesting, unfortunately the experimental performance of the model does not suggest a compelling use case. It is unclear why this model would be preferred over existing RNN models: for longer time series, or in a forecasting setting, there does not seem to be a clear advantage to using the proposed model. For a paper in TMLR, I would like to see a more compelling argument for why and when this model should be preferred to state-of-the-art RNN models, and unfortunately in its present form, this paper does not provide that.

**Audience:**

The idea of constructing RNN cells based on ARMA models is interesting and certainly within the scope of TMLR. However, the performance of the models does not show a significant advantage over alternative methods, and overall, the reader has little motivation to use ARMA cell over alternatives.

**Claims And Evidence:**

The initial submission did not present a compelling case for why ARMA cell should be preferred over alternative methods, both in terms of motivation and in terms of evaluation. The authors did a good job of clarifying the motivation and relationships between models. We appreciate the additional experiments added in response to these comments; however unfortunately they do not show a clear benefit of the proposed model. Moreover, there is still no comparison to state-of-the-art RNN/transformer approaches as suggested by reviewer enze.

---

> ### Author Response · Authors · 2023-02-14
> **Decision inquiry**
>
> Dear Action Editor,
>
> Thank you for your response. We were surprised to see the paper has been rejected on the basis of the reasoning you provided. In particular, you mention that
>
> > [...] there is still no comparison to state-of-the-art RNN/transformer approaches as suggested by reviewer enze.
>
> We **did include a state-of-the-art RNN model** (the DeepAR model) as suggested by reviewer enze. The results can be found in Table 5 of the revised paper version.
>
> Further, you mention that
>
> > [...] the performance of the models does not show a significant advantage over alternative methods [and] the experimental performance of the model does not suggest a compelling use case.
>
> While we think that the model in itself is interesting (as you also mention) and provides a missing connection between classical time series modeling and deep learning, we want to highlight that in fact **our cell significantly outperforms** the LSTM and GRU-cell alternatives on most data sets or is at least not significantly worse
>
> * Table 1: Significantly better in 1 out of 4 experiments; not significantly worse than alternatives
> * Table 2: On par or better in all three experiments
> * Table 3: Significantly better in 2 out of 5 experiments with an error reduction of 33% in one case; not significantly worse else
> * Table 4: Significantly better in 4 out of 8 experiments; significantly worse only in one experiment
> * Table 5: Comparison with state-of-the-art RNN yielding similar results, but our approach is more stable, leading to an error reduction of 50% on Electricity
> * Table 6: Best model in 14 out of 15 cases with a significant error reduction in many cases
>
> Even for longer time series (Tables 7 and 8) and forecasting horizons (Tables 9 and 10), our method is the best of the compared methods in aggregate and provides a strong and stable default method for practitioners and is thus interesting for readers of TMLR.
>
> As a result, we concluded that the idea is not only interesting to the audience of TMLR but also that all of the claims are clearly supported by empirical evidence.
>
> In this light, we would kindly like to ask the AE to reconsider the decision.
>
> Best regards, The Authors